# AN EFFICIENT FRAMEWORK FOR CREDITING DATA CONTRIBUTORS OF DIFFUSION MODELS

**Chris Lin,**\* **Mingyu Lu,**\* **Chanwoo Kim, Su-In Lee**
Paul G. Allen School of Computer Science & Engineering
University of Washington
`{clin25,mingyulu,chanwkim,suinlee}@cs.washington.edu`

## ABSTRACT

As diffusion models are deployed in real-world settings, and their performance is driven by training data, appraising the contribution of data contributors is crucial to creating incentives for sharing quality data and to implementing policies for data compensation. Depending on the use case, model performance corresponds to various global properties of the distribution learned by a diffusion model (e.g., overall aesthetic quality). Hence, here we address the problem of attributing global properties of diffusion models to data contributors. The Shapley value provides a principled approach to valuation by uniquely satisfying game-theoretic axioms of fairness. However, estimating Shapley values for diffusion models is computationally impractical because it requires retraining on many training data subsets corresponding to different contributors and rerunning inference. We introduce a method to efficiently retrain and rerun inference for Shapley value estimation, by leveraging model pruning and fine-tuning. We evaluate the utility of our method with three use cases: (i) image quality for a DDPM trained on a CIFAR dataset, (ii) demographic diversity for an LDM trained on CelebA-HQ, and (iii) aesthetic quality for a Stable Diffusion model LoRA-finetuned on Post-Impressionist artworks. Our results empirically demonstrate that our framework can identify important data contributors across models' global properties, outperforming existing attribution methods for diffusion models[1].

## 1 INTRODUCTION

Diffusion models have demonstrated impressive performance on image generation (Ho et al., 2020; Song et al., 2020b), with models such as Dall·E 2 (Ramesh et al., 2022) and Stable Diffusion (Rombach et al., 2022) showing versatile utilities and enabling downstream applications via customization (Hu et al., 2021; Ruiz et al., 2023). A key driver for the performance of diffusion models is the data used for training and fine-tuning. The data for commercial diffusion models are often scraped from the internet (Schuhmann et al., 2022), raising concerns about credit attribution for those who created the data in the first place (e.g., artists and their artworks) (Jiang et al., 2023). Additionally, as labor is required to label and curate data to further improve model performance, demands for data labeling platforms have risen, with workers often underpaid (Widder et al., 2023; Wong, 2023). To create incentives for sharing quality data and implement policies for data compensation, a pressing question arises: *How do we fairly credit data contributors of diffusion models?*

*Data attribution*, which aims to trace machine learning model behaviors back to training data, has the potential to address the above question. Indeed, in the context of supervised learning, several works have proposed methods for valuating the contribution of individual datum to model performance such as accuracy (Ghorbani & Zou, 2019; Kwon & Zou, 2021; Wang & Jia, 2023). Some recent work has developed data attribution methods for diffusion models (Dai & Gifford, 2023; Georgiev et al., 2023; Wang et al., 2023a; Zheng et al., 2023). However, two gaps remain when applying these recent methods to credit data contributors of diffusion models. First, these methods focus on *local* properties related to the generation of a given image. For example, Zheng et al. (2023) showcase their

---

\*Equal contribution.
[1]Our code is available at `https://github.com/q8888620002/data_attribution`

method D-TRAK to study the changes in pixel values of particular generated images. In contrast, the performance of diffusion models is evaluated based on *global* properties of the learned generative distributions. For example, the demographic diversity of generated images can be considered a global property for evaluation (Luccioni et al., 2023). Second, existing attribution methods for diffusion models consider the contribution of each training datum instead of each data contributor, but a contributor can provide multiple data points. One reasonable approach is to aggregate the valuations of data provided by a contributor as the contributor's total contribution. However, previous work has shown this approach to incur errors between the aggregated and actual contribution (Koh et al., 2019). Here, we aim to address these two gaps by attributing global properties of diffusion models to data contributors.

Attribution methods based on cooperative game theory are particularly desirable because of their axiomatic justification. Specifically, the Shapley value provides a principled approach to fairly distribute credit among contributors, since it is the unique notion that satisfies game-theoretic axioms for equitable valuation (Ghorbani & Zou, 2019; Shapley, 1953). Briefly, the Shapley value assesses each contributor based on the average gain incurred by adding the contributor's data to different contributor combinations. To estimate the Shapley values for data contributors in our setting, we need to (i) retrain diffusion models on data subsets corresponding to different combinations of contributors; and (ii) measure global properties of the retrained models by rerunning inference. However, training a diffusion model can take hundreds of GPU days, and inference can also be expensive (e.g., approximately 5 GPU days to generate 50,000 images for image quality metrics) (Dhariwal & Nichol, 2021). Therefore, estimating Shapley values with vanilla retraining and inference is computationally impractical. Here, we propose to efficiently approximate retraining and inference on retrained models through model pruning and fine-tuning, providing a framework that enables Shapley value estimation (Figure 1).

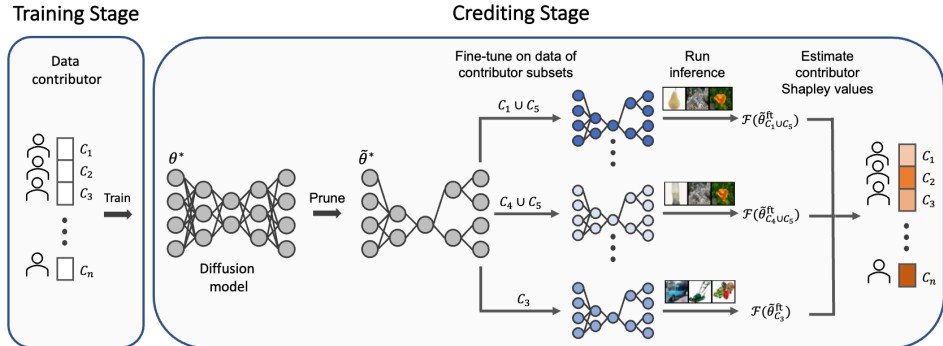

Figure 1: Schematic overview of our proposed framework, where $\theta^*$ denotes a trained diffusion model for which we aim to credit data contributors, and $\tilde{\theta}^*$ denotes the pruned model that approximates $\theta^*$. After fine-tuning the pruned model on data corresponding to various subsets of contributors, denoted as $\tilde{\theta}^{\text{ft}}$, and rerunning inference; global model properties ($\mathcal{F}$) are measured to estimate the Shapley value for each data contributor.

**Related work.** Data attribution methods for diffusion models have been developed by recent work. Some methods require models be trained with specialized paradigms. For example, to assess the importance for a training sample, Dai & Gifford (2023) first train an ensemble of diffusion models on data splits, followed by ablating models trained on splits containing the specific sample. Wang et al. (2023a) evaluate data attribution for text-to-image models by customizing a pretrained model toward an exemplar style. In contrast, other methods can be applied to already trained models in a post hoc manner. For example, the TRAK framework has been adapted to find important training data for intermediate latents along a generative process (Georgiev et al., 2023), while Zheng et al. (2023) introduce empirical approaches that improve the performance of TRAK for diffusion models. All these methods attribute local model properties to each individual datum, whereas our work focuses on attributing global model properties to each data contributor.

**Contributions.** (1) To our knowledge, we are the first to investigate how to attribute global properties of diffusion models to data contributors. (2) We propose a framework that efficiently approximates retraining and rerunning inference for diffusion models, enabling the estimation of Shapley values for

data contributors. (3) We empirically demonstrate that our framework outperforms existing attribution methods across three datasets, model architectures, and global properties.

## 2 PRELIMINARIES

This section provides an overview for diffusion models, attributing global model properties to data contributors, and existing attribution methods for diffusion models.

### 2.1 DIFFUSION MODELS

Our research primarily focuses on discrete-time diffusion models, specifically denoising diffusion probabilistic models (DDPMs) (Ho et al., 2020) and latent diffusion models (LDMs) (Rombach et al., 2022). Generally, diffusion models are trained to approximate a data distribution $q(\mathbf{x}_0)$. To perform learning, a training sample $\mathbf{x}_0 \sim q(\mathbf{x}_0)$ is sequentially corrupted by additive noise (Ho et al., 2020). This procedure is called the *forward process* and is defined by $q(\mathbf{x}_t|\mathbf{x}_{t-1}) := \mathcal{N}(\mathbf{x}_t; \sqrt{1-\beta_t}\mathbf{x}_{t-1}, \beta_t\boldsymbol{I})$, for $t = 1, ..., T$, where $\{\beta_t\}_{t=1}^T$ corresponds to a variance schedule. Notably, the forward process allows sampling of $\mathbf{x}_t$ at any time step $t$ from $\mathbf{x}_0$, with the closed form $q(\mathbf{x}_t|\mathbf{x}_0) = \mathcal{N}(\mathbf{x}_t; \sqrt{\bar{\alpha}_t}\mathbf{x}_0, (1-\bar{\alpha}_t)\boldsymbol{I})$, where $\alpha_t := 1 - \beta_t$ and $\bar{\alpha}_t := \prod_{s=1}^t \alpha_s$. Then, a diffusion model learns to denoise $\mathbf{x}_{1:T}$, following the *reverse process* defined by $p_\theta(\mathbf{x}_{t-1}|\mathbf{x}_t) := \mathcal{N}(\mathbf{x}_{t-1}; \mu_\theta(\mathbf{x}_t, t), \sigma_t^2\boldsymbol{I})$, where $\theta \in \mathbb{R}^d$ is the model parameters, and $\sigma_t$ corresponds to some sampling schedule (Karras et al., 2022). Instead of modeling the conditional means $\mu_\theta$, it is standard to predict the added noises with a neural network $\epsilon_\theta$ using the reparameterization trick. The training objective corresponds to a variational bound and is formulated as

$$\mathcal{L}_{\text{Simple}}(\mathbf{x}; \theta) = \mathbb{E}_{t,\epsilon}\left[\|\epsilon - \epsilon_\theta(\sqrt{\bar{\alpha}_t}\mathbf{x} + \sqrt{(1-\bar{\alpha}_t)}\epsilon, t)\|_2^2\right] \tag{1}$$

Once a diffusion model has been trained, a new image can be generated by sampling an initial noise $\mathbf{x}_T \sim \mathcal{N}(\mathbf{0}, \boldsymbol{I})$ and iteratively applying $\epsilon_\theta$ at each step $t = T, ..., 1$ for denoising.

### 2.2 ATTRIBUTING GLOBAL MODEL PROPERTIES TO DATA CONTRIBUTORS

To gain a comprehensive understanding of machine learning models, it is often beneficial to study global model properties (Covert et al., 2020), which assess the model's performance across samples, rather than focusing on individual outputs. For example, in the setting of supervised learning, global model property can be test accuracy. In the context of generative models, it can be a quality metric for generated samples. For example, in image generation, the global model property of interest can be the Inception Score (Goodfellow et al., 2014) or FID (Heusel et al., 2017). More formally, a global model property for generative models is defined by any application-specific function $\mathcal{F} : \Theta \to \mathbb{R}$, which maps a generative model to a scalar value, thereby quantifying the overall distribution learned by the model. Building on this, we introduce the problem of attributing global model properties to data contributors. The goal of contributor attribution is to identify important groups of training data for a model's global properties, where each group of data is provided by a contributor. More formally, we have the following definition.

**Definition 1.** *(Contributor attribution) Consider $n$ contributors of training samples $\mathcal{D} = \{\mathcal{C}_1, \mathcal{C}_2, ..., \mathcal{C}_n\}$, with the $i$th contributor providing a set of data points denoted as $\mathcal{C}_i$, and a global model property function $\mathcal{F}$. A contributor attribution method is a function $\tau(\mathcal{F}, \{\mathcal{C}_i\}_{i=1}^n)$ that assigns scores to all contributors to indicate each contributor's importance to the global model property $\mathcal{F}$.*

### 2.3 EXISTING ATTRIBUTION METHODS FOR DIFFUSION MODELS

In the context of diffusion models, recent work has focused on attributing local model properties to each datum by adapting the TRAK framework (Park et al., 2023), such as D-TRAK (Zheng et al., 2023) and Journey-TRAK (Georgiev et al., 2023). Formally, let $\mathcal{X}$ denote the input space and $\Theta$ the parameter space of a diffusion model. Suppose there are $N$ training data points $\{\mathbf{x}^{(1)}, \mathbf{x}^{(2)}, ..., \mathbf{x}^{(N)}\}$, and $\tilde{\mathbf{x}}$ is the generated image of interest for local attribution. Given a loss function $\mathcal{L} : \mathcal{X} \times \Theta \to \mathbb{R}$, a local model property $f : \mathcal{X} \times \Theta \to \mathbb{R}$, and models $\{\theta_s^*\}_{s=1}^S$ trained on $S$ data subsets[2], TRAK for

---

[2]In practice, we consider the computationally efficient retraining-free setting by Zheng et al. (2023). That is, $S = 1$, and $\theta_1^* = \theta^*$ is the original model trained on the entire dataset.

diffusion models is defined as:

$$\frac{1}{S} \sum_{s=1}^{S} \Phi_s \left( \Phi_s^\top \Phi_s + \lambda I \right)^{-1} \gamma_s(\tilde{\mathbf{x}}), \text{ and} \tag{2}$$

$$\Phi_s = \left[ \phi_s(\mathbf{x}^{(1)}), \ldots, \phi_s(\mathbf{x}^{(N)}) \right]^\top, \tag{3}$$

where $\phi_s(\mathbf{x}) = \mathcal{P}_s^\top \nabla_\theta \mathcal{L}(\mathbf{x}; \theta_s^*)$, $\gamma_s(\mathbf{x}) = \mathcal{P}_s^\top \nabla_\theta f(\mathbf{x}; \theta_s^*)$, $\mathcal{P}_s$ is a random projection matrix, and $\lambda I$ serves for numerical stability and regularization. More details on Journey-TRAK (Georgiev et al., 2023) and D-TRAK (Zheng et al., 2023) are in Appendix E.

## 3 CREDITING DATA CONTRIBUTORS OF DIFFUSION MODELS USING THE SHAPLEY VALUE

Here, we motivate the use of the Shapley value for crediting data contributors of diffusion models. While estimating Shapley values for diffusion models require computationally expensive retraining and inference, we propose to address the computational challenge with model pruning and fine-tuning.

### 3.1 THE SHAPLEY VALUE FOR CONTRIBUTOR ATTRIBUTION

The Shapley value was developed in cooperative game theory to fairly attribute credit in coalition games (Shapley, 1953). In the context of contributor attribution, for training data with $n$ contributors, the Shapley value attributed to the $i$th contributor is defined as:

$$\beta_i = \frac{1}{n} \sum_{S \subseteq D \backslash \mathcal{C}_i} \binom{n-1}{|S|}^{-1} \left( \mathcal{F}(\theta_{S \cup \mathcal{C}_i}^*) - \mathcal{F}(\theta_S^*) \right) \tag{4}$$

where $\theta_{S \cup \mathcal{C}_i}^*$ and $\theta_S^*$ are models trained on the subsets $S \cup \mathcal{C}_i$ and $S$, respectively. The Shapley value appraises each contributor's contribution based on the weighted *marginal contribution*, $\mathcal{F}(\theta_{S \cup \mathcal{C}_i}^*) - \mathcal{F}(\theta_S^*)$. It satisfies axioms desirable for equitable attribution, including *linearity*, *dummy player*, *symmetry*, and *efficiency* (see Ghorbani & Zou (2019) for a succinct summary). There are arguments that the efficiency axiom, $\sum_{i=1}^{n} \beta_i = \mathcal{F}(\theta^*) - \mathcal{F}(\theta_\emptyset)$, where $\theta^*$ denotes the model trained on the entire dataset, and $\theta_\emptyset$ denotes the model without training, may not always be essential in use cases where the primary goal is to remove detrimental data points (Wang & Jia, 2023). In that setting, ranking is more important than exact attribution values. However, in the context of diffusion models, particularly in applications involving monetary or credit allocation related to model performance, having the total attribution matches the model utility can directly quantify reward for each contributor.

Despite its advantages, evaluating the *exact* Shapley value is challenging as it involves retraining $2^n$ (all possible subsets) models and computing their corresponding model properties. Fortunately, many sampling-based estimators for Shapley values have been developed (Lundberg & Lee, 2017; Ghorbani & Zou, 2019; Štrumbelj & Kononenko, 2010). In particular, we adopt KernelSHAP in the feature attribution literature for contributor attribution, by solving a weighted least squares problem with sampling (Lundberg & Lee, 2017; Covert & Lee, 2021):

$$\hat{\beta} = \min_{\beta_0, \ldots, \beta_n} \frac{1}{M} \sum_{j=1}^{M} \left[ \mathcal{F}(\theta_\emptyset) + \mathbf{1}_{S_j}^\top \beta - \mathcal{F}(\theta_{S_j}^*) \right]^2 \quad \text{s.t.} \quad \mathbf{1}^\top \beta = \mathcal{F}(\theta^*) - \mathcal{F}(\theta_\emptyset) \tag{5}$$

where $S_j$ is sampled following the distribution $\mu(S) \propto \frac{n-1}{\binom{n}{|S|}|S|(n-|S|)}$ for $1 < \mathbf{1}_S^\top \beta < n$, and $\mathbf{1}_S$ is an indicator vector representing the presence of contributors in $S$. By solving the least squares problem with the constraint, a closed-form solution can be derived (Covert & Lee, 2021). The obtained parameters $\hat{\beta}_1, \ldots, \hat{\beta}_n$ are the contributor attribution scores.

### 3.2 SPEED UP RETRAINING AND INFERENCE WITH SPARSIFIED FINE-TUNING

Although using the Shapley value for contributor attribution is well motivated, and existing sampling approach can be used, the main challenge is in computing $\mathcal{F}(\theta_{S_j}^*)$. First, a diffusion model needs to

be trained again from random initialization on the subset $S_j$ to obtain $\theta_{S_j}^*$. Then inference needs to be rerun on the retrained model $\theta_{S_j}^*$ to measure $\mathcal{F}(\theta_{S_j}^*)$. For diffusion models, both retraining and rerunning inference can take multiple GPU days (Dhariwal & Nichol, 2021), making the estimator in Equation (5) computationally impractical.

To address this computational challenge, we propose to speed up retraining and rerunning inference with *sparsified fine-tuning* (Figure 1). Sparsified fine-tuning enhances the efficiency of both retraining and inference, by reducing the number of model parameters through pruning (Jia et al., 2023). Specifically, $\theta^*$ is pruned and initially fine-tuned on the full dataset $\mathcal{D}$ to obtain a performant pruned model $\tilde{\theta}^*$ that approximates $\theta^*$. To furthermore improve the retraining efficiency, $\tilde{\theta}^*$ is fine-tuned on each subset $S_j$ for $k$ steps to obtain $\tilde{\theta}_{S_j,k}^{\text{ft}}$, instead of retraining the pruned model on $S_j$ from random initialization. The diffusion loss objective used to train the original model $\theta^*$ is used for all fine-tuning. Overall, sparsified fine-tuning aims to efficiently achieve

$$\mathcal{F}(\tilde{\theta}_{S_j,k}^{\text{ft}}) \ (\textit{sparsified fine-tuning}) \approx \mathcal{F}(\theta_{S_j}^*) \ (\textit{retraining full model from scratch}). \tag{6}$$

In other words, under the same computational budget and compared to retaining the full model from scratch, sparsified fine-tuning can increase the number of sampled subsets $S_j$ for estimating Shapley values, thus making feasible a framework for crediting data contributors of diffusion models.

In practice, computing $\mathcal{F}(\theta)$ requires sampling $N$ initial noises $\mathbf{x}_T^{(r)} \sim \mathcal{N}(\mathbf{0}, \mathbf{I})$, for $r = 1, ..., N$, denoising those noises into generated samples with the denoising network $\epsilon_\theta$, and computing some quantity over the generated samples (e.g., FID). Hence, to formally analyze the approximation in Equation (6), we consider the expected error[3] $\mathbb{E}_{\{\mathbf{x}_T^{(r)}\}_{r=1}^N}[|\mathcal{F}(\tilde{\theta}_{S_j,k}^{\text{ft}}) - \mathcal{F}(\theta_{S_j}^*)|]$. For notational ease, the dependency of $\mathcal{F}$ on $\{\mathbf{x}_T^{(r)}\}_{r=1}^N$ is omitted, and all expectations are taken with respect to $\{\mathbf{x}_T^{(r)}\}_{r=1}^N$ for the rest of the paper unless otherwise noted. We then have the following proposition to formalize the intuition behind Equation (6).

**Proposition 1.** *Suppose an objective function $\ell : \mathbb{R}^d \mapsto \mathbb{R}$ on the data provided by a given subset of data contributors $S$ is convex and differentiable, and that its gradient is Lipschitz-continuous with some constant $L > 0$, i.e.*

$$\|\nabla \ell(\theta_1) - \nabla \ell(\theta_2)\|_2 \leq L \|\theta_1 - \theta_2\|_2$$

*for any $\theta_1, \theta_2 \in \mathbb{R}^d$. Let $\tilde{\theta}_{S,k}^{ft}$ denote a pruned model after $k$ fine-tuning steps on the given subset $S$ with learning rate $\alpha \leq 1/L$, $\tilde{\theta}_S^*$ the optimal pruned model trained on $S$ with the same sparsity structure as $\tilde{\theta}_{S,k}^{ft}$, and $\theta_S^*$ the optimal full-parameter model trained on $S$. Furthermore, assume that $\mathbb{E}[|\mathcal{F}(\tilde{\theta}_S^*) - \mathcal{F}(\theta_S^*)|] \leq B$ for some constant $B$. Then the expected error $\mathbb{E}[|\mathcal{F}(\tilde{\theta}_{S,k}^{ft}) - \mathcal{F}(\theta_S^*)|] \leq B$ as $k \to \infty$.*

The proof is in Appendix A. The assumption that $\ell$ is convex and differentiable with Lipschitz-continuous gradient is a standard setting for the theoretical analysis of approximating retraining with fine-tuning, instantiated as a quadratic objective[4] in Golatkar et al. (2020) and Georgiev et al. (2024). The assumption that $\mathbb{E}[|\mathcal{F}(\tilde{\theta}_S^*) - \mathcal{F}(\theta_S^*)|] \leq B$ corresponds to the *lottery ticket hypothesis* that a sparser neural network can approximate the performance of a dense, full-parameter network (Frankle & Carbin, 2018), which has been shown to hold for diffusion models (Fang et al., 2023). The main takeaway from Proposition 1 is that more steps of sparsified fine-tuning should lead to a bounded approximation error between $\mathcal{F}(\tilde{\theta}_{S_j,k}^{\text{ft}})$ and $\mathcal{F}(\theta_{S_j}^*)$ in Equation (6). We further relate Proposition 1 to the expected error in Shapley values with the following proposition.

**Proposition 2.** *Let $\tilde{\beta}_k^{ft} \in \mathbb{R}^n$ be the Shapley values for the data contributors evaluated with $\{\mathcal{F}(\tilde{\theta}_{S,k}^{ft})\}_{S \in 2^{\mathcal{D}}}$, and $\beta^* \in \mathbb{R}^n$ the Shapley values evaluated with $\{\mathcal{F}(\theta_S^*)\}_{S \in 2^{\mathcal{D}}}$. Suppose the assumptions on $\ell$ in Proposition 1 hold for all subsets of data contributors. Furthermore, assume that $\mathbb{E}[\max_{S \in 2^{\mathcal{D}}} |\mathcal{F}(\tilde{\theta}_S^*) - \mathcal{F}(\theta_S^*)|] \leq C$. Then $\mathbb{E}[\|\tilde{\beta}_k^{ft} - \beta^*\|_2] \leq 2\sqrt{n}C$ as $k \to \infty$.*

---

[3]We consider the setting where the same $\{\mathbf{x}_T^{(r)}\}_{r=1}^N$ are inputs for both $\epsilon_{\tilde{\theta}_{S_j,k}^{\text{ft}}}$ and $\epsilon_{\theta_{S_j}^*}$ to represent the use case of generating samples with the same random seed across multiple inference runs.

[4]A quadratic objective has the form $\ell(\theta) = (\theta - B)^\top A (\theta - B)$, with Lipschitz-continuous gradient such that $\|\nabla \ell(\theta_1) - \nabla \ell(\theta_2)\|_2 \leq 2\sigma_{\max}(A) \cdot \|\theta_1 - \theta_2\|_2$, where $\sigma_{\max}(A)$ is the maximum singular value of $A$.

The proof is in Appendix A. The assumption that convexity, differentiability, and the Lipschitz continuity of gradient hold for all subsets is implied by the standard setting of a quadratic loss[5], as in Golatkar et al. (2020) and Georgiev et al. (2024). The key takeaway is that the Shapley value error is bounded with more steps of sparsified fine-tuning. Finally, we note that both Proposition 1 and Proposition 2 are asymptotic results, and that Proposition 2 is based on exact Shapley values. We leave theoretical results incorporating finite-step bounds and Shapley value estimation for future work. Nevertheless, we verify the insights from our theoretical results under empirical settings for diffusion models in Appendix D.

## 4 EXPERIMENTS

In this section, we compare our approach with existing attribution methods across three settings. We employ two metrics for evaluation: linear datamodeling score (LDS) and counterfactual analysis. Our results demonstrate that our method outperforms existing attribution methods for diffusion models.

### 4.1 DATASETS AND CONTRIBUTORS

Datasets used for our experiments include CIFAR-20, a subset of CIFAR-100 with 20 contributors, one per class, as described in Krizhevsky et al. (2009); CelebA-HQ, with 50 celebrity identities as contributors; and ArtBench (Post-Impressionism), with 258 artists as contributors. More details about datasets and contributors can be found in Appendix B.

### 4.2 EXPERIMENT SETTINGS

**Model training.** For CIFAR-20, we follow the original implementation of the unconditional DDPM (Ho et al., 2020) where the model has 35.7M parameters. For CelebA-HQ, we follow the implementation of LDM (Rombach et al., 2022) with 274M parameters and a pre-trained VQ-VAE (Razavi et al., 2019). For ArtBench (Post-Impressionism), a Stable Diffusion model (Rombach et al., 2022) is fine-tuned using LoRA (Hu et al., 2021) with rank = 256, corresponding to 5.1M LoRA parameters. The prompt for the Stable Diffusion model is set to *"a Post-Impressionist painting"* for each image. Please refer to Appendix C for more details about model training and inference.

**Sparsified fine-tuning.** Magnitude-based pruning (Han et al., 2015) is used to remove model weights according to their magnitudes, resulting in sparse diffusion models with reduced parameters: from 35.7M to 19.8M for CIFAR-20, from 274M to 70.9M for CelebA-HQ, and from 5.1M to 2.6M for ArtBench (Post-Impressionism). To estimate Shapley values, the pruned models are fine-tuned on data subsets corresponding to different contributor combinations, with 1,000 steps, 500 steps, and 200 steps for CIFAR-20, CelebA-HQ, and ArtBench (Post-Impressionism), respectively.

**Global model properties.** For CIFAR-20, we aim to study the contribution of each labeler. As each labeler is tasked to filter out noisy samples in each class (Krizhevsky et al., 2009), high-quality labeling should ensure that generated images are well separated with respect to image classes. Therefore, for the global model property, we choose the Inception Score (Salimans et al., 2016):

$$\text{IS} = \exp\left(\mathbb{E}_{\mathbf{x}}[\text{KL}(p(y|\mathbf{x})\|p(y))]\right), \tag{7}$$

where $p(y)$ represents the marginal class distribution over the generated data and $p(y|\mathbf{x})$ represents the conditional class distribution given a generated image $\mathbf{x}$.

For CelebA-HQ, our goal is to investigate how individual celebrities contribute to the demographic diversity of the model. Following Luccioni et al. (2023), we measure diversity using entropy:

$$\mathcal{H} = -\sum_{k=1}^{K} p_k \log(p_k), \tag{8}$$

where $p_k$ is the proportion of generated samples in the $k$th demographic cluster. We generate images using the full model and extract their embeddings with BLIP-VQA (Li et al., 2022) to create 20

---

[5]A Lipschitz constant applicable across all subsets can be defined as $\max_{S \in 2^{\mathcal{D}}} \sigma_{\max}(A_S)$, where $A_S$ is defined by the subset $S$. For example, in linear regression, $A_S = X_S^\top X_S$, where $X_S$ denotes the feature matrix contributed by the data contributors in $S$.

reference clusters. Subsequently, we assign these reference clusters to images generated from various retrained models to calculate the entropy.

For ArtBench (Post-Impressionism), we consider the use case where images are generated and the most aesthetically pleasing ones are kept. We simulate this use case by computing aesthetic scores[6] for 50 generated images and considering the 90th percentile as the global model property.

More details about these global model properties are in Appendix B.

### 4.3 BASELINE METHODS

We mainly categorize baseline attribution methods into three categories[7]: (i) similarity-based methods (e.g., pixel similarity between the generated and training images); (ii) leave-one-out (LOO) and its approximate variants such as the influence functions (Koh & Liang, 2017) and TRAK-based methods (Park et al., 2023); and (iii) application-specific metrics such as the aesthetic score of each training image for ArtBench (Post-Impressionism). Because retraining diffusion models is computationally expensive, gradients are computed using the original model $\theta^*$ for TRAK-based methods, as done in Zheng et al. (2023). Local attribution scores are averaged across the images generated by the original model for computing global model properties. To aggregate datum-level attributions, the sum of attributions corresponding to each contributor is taken for the influence functions and TRAK-based methods, since this is the principled aggregation (Koh et al., 2019). For similarity-based and application-specific methods, datum-level attribution scores are aggregated for contributor attribution by taking either the average or maximum. More details about baseline methods can be found in Appendix E.

### 4.4 EVALUATING THE PERFORMANCE OF CONTRIBUTOR ATTRIBUTION

**Linear datamodeling score (LDS).** From Definition 1, for a given subset of contributors $S \subseteq \mathcal{D}$, where $\mathcal{D} = \{\mathcal{C}_1, ..., \mathcal{C}_n\}$, we can define an additive datamodel of global properties based on a given set of attribution scores $\tau$ for the contributors:

$$g(S, \tau) = \sum_{i:\mathcal{C}_i \in S} \tau_i. \tag{9}$$

Therefore, the evaluation for an attribution method $\tau$ can be constructed as follows:

**Definition 2.** *(Linear datamodeling score) Contributor attribution performance is measured using the linear datamodeling score (LDS) (Ilyas et al., 2022), which evaluates an attribution method by comparing predicted model properties based on the additive datamodel against actual retrained model properties. Let $S_1, ..., S_B$ be randomly sampled subsets of $\mathcal{D}$, each of size $\alpha \cdot n$ for some $\alpha \in (0, 1)$. The LDS for a contributor attribution score $\tau \in \mathbb{R}^n$ is defined as*

$$LDS = \rho\left(\{\mathcal{F}(\theta^*_{S_b})\}_{b=1}^B, \{g(S_b, \tau)\}_{b=1}^B\right), \text{ where } S_b \sim Uniform\{S \subset \mathcal{D} : |S| = \alpha \cdot n\}, \tag{10}$$

*where $\rho$ is the Spearman rank correlation (Spearman, 1961), and $\theta^*_{S_b}$ denotes a model retrained from scratch with the contributor subset $S_b$.*

We evaluate the LDS using 100 held-out subsets $S_b$, each sampled from the datamodel distribution with $\alpha = 0.25, 0.5, 0.75$ for each dataset. We report the LDS means and 95% confidence intervals across three independent sets of $\{S_b\}_{b=1}^{100}$.

**Counterfactual evaluation.** Following Zheng et al. (2023), we also apply counterfactual evaluation (Hooker et al., 2019). We assess the relative change in model property, $\Delta\mathcal{F} = \frac{\mathcal{F}(\theta^*_K) - \mathcal{F}(\theta^*)}{\mathcal{F}(\theta^*)}$, by comparing the models trained before and after excluding (or retaining only) the top $K$ most influential contributors identified by each attribution method. Counterfactual evaluation requires retraining models on different subsets for each method, so only baseline methods with the best LDS in each category are chosen for computational feasibility.

---

[6]https://github.com/LAION-AI/aesthetic-predictor
[7]TracIn (Pruthi et al., 2020) is not considered since intermediate checkpoints may not be available in practice.

### 4.5 Experiment Results

#### Shapley Attribution Outperforms Baseline Methods in Contributor Attribution

In Table 1, we present the LDS results for baseline methods and our approach. Interestingly, similarity-based methods such as raw pixel similarity, CLIP similarity, and gradient similarity occasionally outperform TRAK-based methods. We observe that TRAK-based methods sometimes yield poor or even negative correlations. Among TRAK-based approaches, attribution using noisy latents during generation (i.e., Journey-TRAK) can result in a negative LDS for model properties such as the Inception Score and aesthetic score. Our findings indicate that simply aggregating individual attributions derived from diffusion loss or its alternative functions is insufficient for accurately determining contributor attribution of global properties. Such an approach can perform worse than simple heuristics based on model properties (e.g., the average aesthetic score).

Table 1: LDS (%) results with $\alpha = 0.5$. Means and 95% confidence intervals across three random initializations are reported.

| Method | CIFAR-20 | CelebA-HQ | ArtBench (Post-Impressionism) |
|---|---|---|---|
| Pixel similarity (average) | -11.81 ± 4.56 | -8.91 ± 0.93 | 11.24 ± 0.63 |
| Pixel similarity (max) | -31.80 ± 2.90 | 21.70 ± 2.05 | 14.61 ± 2.72 |
| Embedding dist. (average) | - | 13.83 ± 1.12 | - |
| Embedding dist. (max) | - | 7.32 ± 3.16 | - |
| CLIP similarity (average) | 5.79 ± 3.67 | -32.23 ± 0.87 | -6.96 ± 4.08 |
| CLIP similarity (max) | 11.31 ± 0.37 | -0.93 ± 3.83 | -1.75 ± 4.07 |
| Gradient similarity (average) | 5.79 ± 3.67 | -18.32 ± 0.65 | 0.25 ± 1.18 |
| Gradient similarity (max) | -0.89 ± 3.17 | -12.90 ± 1.60 | 10.48 ± 3.11 |
| Aesthetic score (average) | - | - | 24.85 ± 2.30 |
| Aesthetic score (max) | - | - | 21.36 ± 3.70 |
| Relative IF | 5.23 ± 5.50 | -1.07 ± 0.68 | -5.02 ± 1.77 |
| Renormalized IF | 11.39 ± 6.79 | 10.17 ± 0.57 | -11.41 ± 0.93 |
| TRAK | 7.94 ± 5.67 | 3.22 ± 0.75 | -8.18 ± 1.30 |
| Journey-TRAK | -42.92 ± 2.15 | -2.88 ± 4.02 | -11.41 ± 4.22 |
| D-TRAK | 10.90 ± 1.21 | −27.23 ± 2.80 | 11.30 ± 3.47 |
| Leave-one-out (LOO) | 30.66 ± 6.11 | -1.22 ± 6.34 | 3.74 ± 8.00 |
| Sparsified-FT Shapley (**Ours**) | **61.48 ± 2.27** | **26.34 ± 3.42** | **61.44 ± 2.04** |

In contrast, our approach, sparsified fine-tuning (sparsified-FT) Shapley, computes contributor attribution using the Shapley value, achieving the highest LDS results of 61.48%, 26.34%, and 61.44% for CIFAR-20, CelebA-HQ, and ArtBench (Post-Impressionism), respectively. While leave-one-out (LOO) achieves 30.66% LDS on CIFAR-20, its performance declines as the number of contributors increase, e.g., CelebA-HQ and ArtBench (Post-Impressionism). This shows that attribution based on the marginal contribution of Shapley subsets with respect to $\mathcal{F}$ provides the most accurate importance score. Despite achieving the best results for CelebA-HQ compared to the baseline methods, we observe that the LDS performance of sparsified-FT Shapley, 26.32%, is relatively low compared to those of CIFAR-20 and ArtBench. We also perform evaluation with additional datamodel subset sizes ($\alpha = 0.25, 0.75$), and our approach consistently outperforms others (Appendix F).

#### Enhancing Retraining and Inference Efficiency through Sparsified Fine-Tuning

As described in Section 3.2, computing $\mathcal{F}(\theta_{S_j}^*)$ is the primary computational bottleneck. Our sparsified fine-tuning approach significantly reduces the runtime required to obtain $\theta_{S_j}^*$ compared to retraining and fine-tuning without sparsification. On average, retraining and inference with sparsified fine-tuning for a Shapley subset take 18.3 minutes for CIFAR-20, 22.9 minutes for CelebA-HQ, and 10.5 minutes for ArtBench (Post-Impressionism), making it 5.3, 10.4, and 18.6 times faster than retraining, respectively (Table 2). By enabling faster computation and obtaining more models retrained on different subsets, sparsified FT yields the best LDS results under the same computational budgets (Figure 2). This demonstrates that sparsified-FT Shapley is both more computationally

feasible and accurate with limited computational resources. To the best of our knowledge, we are the first to overcome the computational bottleneck and enable contributor attribution using Shapley values for diffusion models.

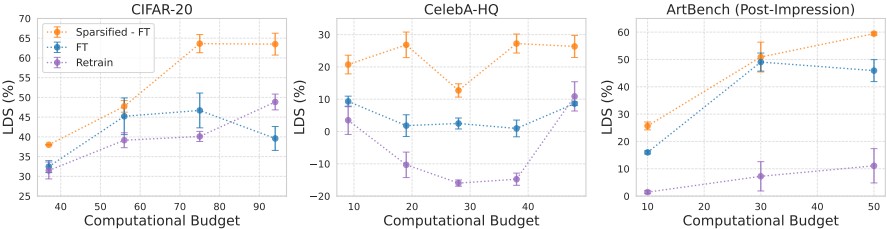

Figure 2: Comparison of LDS (%) with $\alpha = 0.5$ among Shapley values estimated with sparsified fine-tuning (FT), fine-tuning (FT), and retraining under the same computational budgets (1 unit = runtime to retrain and run inference on a full model). Specific runtimes are shown in Table 2.

#### IMPACT OF CONTRIBUTORS ON MODEL PROPERTY

Here we present the results of our counterfactual evaluation, analyzing changes in model behavior after either removing the top contributors or including only the top contributors identified by each method. In CIFAR-20, our approach shows a change of -23.23%, compared to -14.95% for CLIP similarity and -17.30% for LOO, as shown in Figure 3 (top). For CelebA-HQ, the changes are -7.83% for our method, -6.64% for pixel similarity, and 0.21% for renormalized IF. For ArtBench (Post-Impressionism), the changes are 0.58%, -0.05%, -1.27%, and -1.86% for D-TRAK, maximum pixel similarity, average aesthetic score, and sparsified-FT Shapley, respectively.

Conversely, when retaining the top 60% of contributors, model properties improve. In the CIFAR-20 dataset, our method leads to a 16.98% positive change in model properties, compared to -9.45% for CLIP similarity and 9.51% for LOO. For CelebA-HQ, the model behavior change was 20.0% for our method, 8.89% for pixel similarity, and 6.02% for renormalized IF (bottom of Figure 3). These findings, along with the results in Section 4.5, demonstrate that our approach effectively identifies the top contributors and provides accurate attribution.

#### WHO ARE THE TOP CONTRIBUTORS?

We analyze the top contributors identified by sparsified-FT Shapley in CIFAR-20, including contributors to classes such as motorcycles, buses, and lawnmowers (Figure 4). High-quality labeling by these contributors should ensure that the training images contain clear and meaningful objects, which should result in low entropy (i.e., high confidence) of a classifier (i.e., Inception v3) (Barratt & Sharma, 2018). We therefore compute the entropy of each image and find that the top 20% classes have a lower average entropy of 5.18, compared to 6.03 for the remaining classes (Figure 16). For CelebA-HQ, the most important celebrities ranked by sparsified-FT Shapley encompass a diverse range of demographics and tend to belong to non-majority demographic clusters (Figure 16). Excluding images corresponding to these celebrities can have a negative impact on the diversity score. For ArtBench, we observe that images from the top contributors are more vivid and exhibit more vibrant colors, as shown in Figure 4.

## 5 DISCUSSION

In this work, we introduce the problem of attributing global model properties to data contributors of diffusion models. We develop an efficient framework to estimate the Shapley values for data contributors by leveraging model pruning and fine-tuning, speeding up retraining and inference runtime while ensuring accurate attribution. Our framework can have a range of implications, such as creating incentives for data contributors, rewarding data labelers in a way that satisfies game-theoretic fairness, assessing label quality, and improving model performance and fairness. Empirical results for multiple datasets and global model properties show that our framework outperforms existing attribution approaches based on the diffusion loss, such as TRAK (Park et al., 2023) and its variant

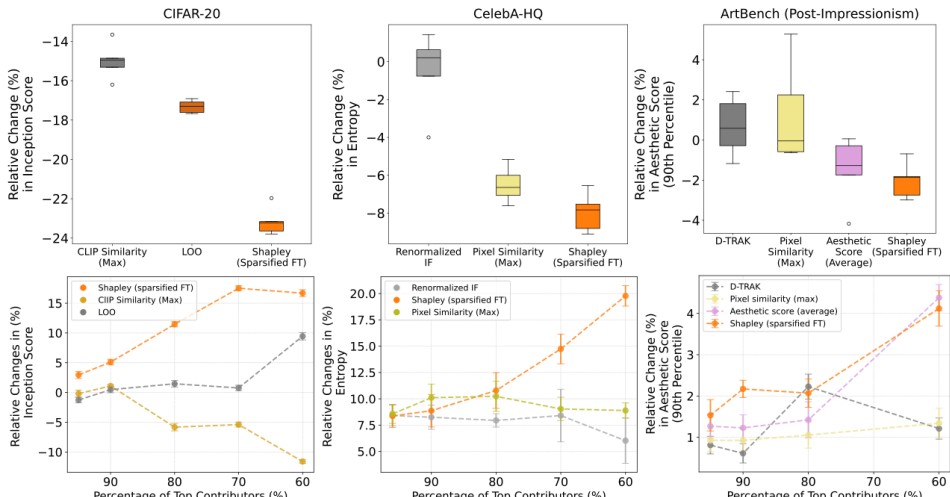

Figure 3: Relative percentage changes in global properties, comparing the original fully trained models to models retrained after removing the top 40% contributors (top) and including only the top contributors (bottom), as identified by various attribution methods.

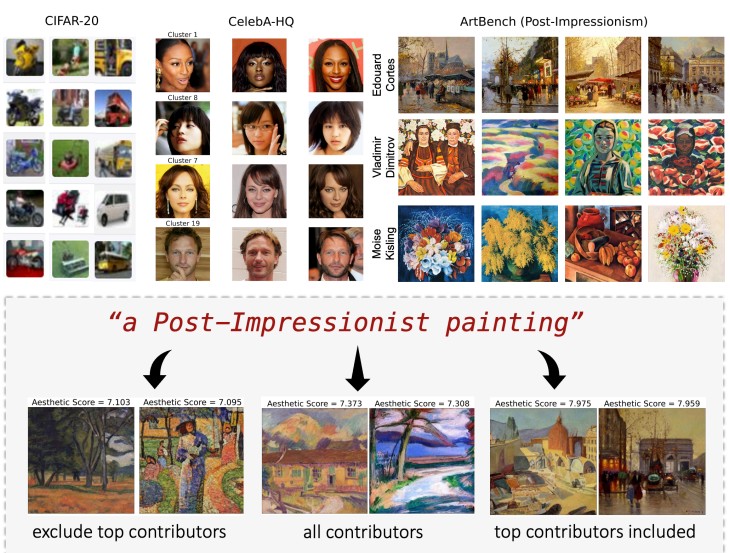

Figure 4: Top contributors and their corresponding training images for each dataset (top). Pairs of generated images above and below the 90th percentile of aesthetic score from Stable Diffusion models LoRA-finetuned under three conditions: excluding the data from the top 40% of artists, using the data from all the artists, and including only the data from the top 60% of artists (bottom).

D-TRAK (Zheng et al., 2023), in crediting data contributors. There are several promising directions for future research. Beyond fine-tuning, recent unlearning methods designed for diffusion models (Gandikota et al., 2023; Heng & Soh, 2023) could be explored in combination with various pruning strategies (Ding et al., 2019; Fang et al., 2024; Liu et al., 2021). Beyond diffusion models, our proposed framework can also be extended to models that are expensive to retrain, such as large language models (LLMs), whenever appropriate pruning and unlearning approaches are available. Finally, our approach and existing attribution methods for diffusion models assume access to training data and model parameters, which are not necessarily available for deployed models. For text-to-image models, membership inference and in-context learning may be useful in addressing this challenge (Hu & Pang, 2023; Wang et al., 2023b).

## 6 ETHICS STATEMENT

As diffusion models become increasingly deployed in real-world applications, there is a notable lack of clear guidelines for crediting data contributors. This work focuses on the fair attribution of data contributors for diffusion models, which is a crucial step towards creating incentives for sharing quality data and implementing policies for equitable compensation. By efficiently estimating Shapley values, our approach can promote fairness in recognizing the importance of data contributors. Importantly, our approach is motivated by positive attribution, such as incentivizing quality labeling and improving the demographic diversity of generated images. However, we also recognize that our approach could potentially be used to remove data from certain data contributors. We view this as an opportunity to ensure higher data quality and transparency in model development, particularly when addressing harmful or irrelevant contributions. Ensuring data contributors are fairly compensated and informed is critical to avoid exploitation. Our use cases, such as demographic diversity for CelebA-HQ, highlight the need for transparency in data usage and advocate for diverse and representative datasets. At the same time, we acknowledge that considerations related to data privacy, consent, and the potential amplification of biases, especially in socially sensitive applications, may not be easily quantified as model properties.

## 7 ACKNOWLEDGEMENTS

We thank members of the Lee lab for providing feedback on this project. This work was funded by the National Science Foundation [DBI-1759487]; and the National Institutes of Health [R01 AG061132, R01 EB035934, and RF1 AG088824].

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

APPENDIX

## A PROOFS

Before proving Proposition 1, we first provide the following lemma.

**Lemma 1.** *Suppose an objective function $\ell : \mathbb{R}^d \mapsto \mathbb{R}$ is convex and differentiable, and that its gradient is Lipschitz-continuous with some constant $L > 0$, i.e.*

$$\|\nabla\ell(\theta_1) - \nabla\ell(\theta_2)\|_2 \le L\|\theta_1 - \theta_2\|_2$$

*for any $\theta_1, \theta_2 \in \mathbb{R}^d$. Then running gradient descent for $k$ steps with learning rate $\alpha \le 1/L$ satisfies*

$$\ell(\theta^{(k)}) - \ell(\theta^*) \le \frac{\|\theta^{(0)} - \theta^*\|_2^2}{2k\alpha},$$

*where $\theta^{(k)}$ is the model parameter after the kth step, $\theta^*$ is the optimum, and $\theta^{(0)}$ is the initial parameter.*

*Proof.* We provide the proof for this well-known result for completeness. The assumption that $\nabla\ell(\theta)$ is $L$-Lipschitz continuous implies that $\nabla^2\ell(\theta) \preceq LI$, so we can perform a quadratic expansion of $\ell$ around $\ell(\theta)$ to obtain the following inequality:

$$\ell(\theta^{(i)}) \le \ell(\theta) + \nabla\ell(\theta)^\top(\theta^{(i)} - \theta) + \frac{1}{2}L\|\theta^{(i)} - \theta\|_2^2, \tag{11}$$

for any $\theta, \theta^{(i)} \in \mathbb{R}^d$. With the gradient descent update $\theta^{(i)} = \theta^{(i-1)} - \alpha\nabla\ell(\theta^{(i-1)})$ and setting $\theta = \theta^{(i-1)}$ in Equation (11), we obtain

$$\ell(\theta^{(i)}) \le \ell(\theta^{(i-1)}) + \nabla\ell(\theta^{(i-1)})^\top\alpha\nabla\ell(\theta^{(i-1)}) + \frac{1}{2}L\|\alpha\nabla\ell(\theta^{(i-1)})\|_2^2 \tag{12}$$

$$= \ell(\theta^{(i-1)}) - \alpha\|\nabla\ell(\theta^{(i-1)})\|_2^2 + \frac{1}{2}\alpha^2 L\|\nabla\ell(\theta^{(i-1)})\|_2^2 \tag{13}$$

$$= \ell(\theta^{(i-1)}) - \left(1 - \frac{1}{2}\alpha L\right)\alpha\|\nabla\ell(\theta^{(i-1)})\|_2^2. \tag{14}$$

Using $\alpha \le 1/L$, we have

$$1 - \frac{1}{2}\alpha L \ge 1 - \frac{1}{2} \cdot \frac{1}{L} \cdot L = \frac{1}{2}. \tag{15}$$

Hence

$$\ell(\theta^{(i)}) \le \ell(\theta^{(i-1)}) - \frac{1}{2}\alpha\|\nabla\ell(\theta^{(i-1)})\|_2^2, \tag{16}$$

which implies that the objective is decreasing until the optimal value is reached, because $\|\nabla\ell(\theta^{(i-1)})\|_2^2$ is positive unless $\nabla\ell(\theta^{(i-1)}) = 0$.

Now since $\ell$ is convex, we have

$$\ell(\theta^{(i-1)}) + \nabla\ell(\theta^{(i-1)})^\top(\theta^* - \theta^{(i-1)}) \le \ell(\theta^*), \tag{17}$$

and

$$\ell(\theta^{(i-1)}) \le \ell(\theta^*) + \nabla\ell(\theta^{(i-1)})^\top(\theta^{(i-1)} - \theta^*) \tag{18}$$

after rearrangement. Plugging this inequality into Equation (16), we have

$$\ell(\theta^{(i)}) \le \ell(\theta^*) + \nabla\ell(\theta^{(i-1)})^\top(\theta^{(i-1)} - \theta^*) - \frac{1}{2}\alpha\|\nabla\ell(\theta^{(i-1)})\|_2^2, \tag{19}$$

which implies that

$$\ell(\theta^{(i)}) - \ell(\theta^*) \tag{20}$$

$$\le \nabla\ell(\theta^{(i-1)})^\top(\theta^{(i-1)} - \theta^*) - \frac{1}{2}\alpha\|\nabla\ell(\theta^{(i-1)})\|_2^2 \tag{21}$$

$$= \frac{1}{2\alpha}\left(2\alpha\nabla\ell(\theta^{(i-1)})^\top(\theta^{(i-1)} - \theta^*) - \alpha^2\|\nabla\ell(\theta^{(i-1)})\|_2^2\right) \tag{22}$$

$$= \frac{1}{2\alpha}\left(2\alpha\nabla\ell(\theta^{(i-1)})^\top(\theta^{(i-1)} - \theta^*) - \alpha^2\|\nabla\ell(\theta^{(i-1)})\|_2^2 - \|\theta^{(i-1)} - \theta^*\|_2^2 + \|\theta^{(i-1)} - \theta^*\|_2^2\right) \tag{23}$$

$$= \frac{1}{2\alpha}\left(\|\theta^{(i-1)} - \theta^*\|_2^2 - \|\theta^{(i-1)} - \alpha\nabla\ell(\theta^{(i-1)}) - \theta^*\|_2^2\right). \tag{24}$$

Plugging in the gradient descent update rule into this inequality, we get

$$\ell(\theta^{(i)}) - \ell(\theta^*) \leq \frac{1}{2\alpha} \left( \|\theta^{(i-1)} - \theta^*\|_2^2 - \|\theta^{(i)} - \theta^*\|_2^2 \right). \tag{25}$$

Summing over $k$ steps, we have

$$\sum_{i=1}^{k} \ell(\theta^{(i)}) - \ell(\theta^*) \leq \sum_{i=1}^{k} \frac{1}{2\alpha} \left( \|\theta^{(i-1)} - \theta^*\|_2^2 - \|\theta^{(i)} - \theta^*\|_2^2 \right) \tag{26}$$

$$= \frac{1}{2\alpha} \left( \|\theta^{(0)} - \theta^*\|_2^2 - \|\theta^{(k)} - \theta^*\|_2^2 \right) \tag{27}$$

$$\leq \frac{1}{2\alpha} \|\theta^{(0)} - \theta^*\|_2^2 \tag{28}$$

, where the telescoping sum results in the equality. Finally, the term in the summation on the LHS is decreasing because of Equation (16), hence

$$\ell(\theta^{(k)}) - \ell(\theta^*) \leq \frac{1}{k} \sum_{i=1}^{k} \ell(\theta^{(i)}) - \ell(\theta^*) \leq \frac{\|\theta^{(0)} - \theta^*\|_2^2}{2k\alpha}. \tag{29}$$

$\square$

We then proceed to prove Proposition 1.

*Proof of Proposition 1.* By the triangle inequality, we have

$$\mathbb{E}[|\mathcal{F}(\tilde{\theta}_{S,k}^{\text{ft}}) - \mathcal{F}(\theta_S^*)|] \leq \mathbb{E}[|\mathcal{F}(\tilde{\theta}_{S,k}^{\text{ft}}) - \mathcal{F}(\tilde{\theta}_S^*)|] + \mathbb{E}[|\mathcal{F}(\tilde{\theta}_S^*) - \mathcal{F}(\theta_S^*)|] \tag{30}$$

$$\leq \mathbb{E}[|\mathcal{F}(\tilde{\theta}_{S,k}^{\text{ft}}) - \mathcal{F}(\tilde{\theta}_S^*)|] + B, \tag{31}$$

where the second inequality comes from the assumption that $\mathbb{E}[|\mathcal{F}(\tilde{\theta}_S^*) - \mathcal{F}(\theta_S^*)|] \leq B$.

Note that $\tilde{\theta}_{S,k}^{\text{ft}}$ and $\tilde{\theta}_S^*$ are in $\mathbb{R}^d$, with zeros for the pruned weights. Hence the assumptions on $\ell$ hold for $\tilde{\theta}_{S,k}^{\text{ft}}, \tilde{\theta}_S^*$. Applying Lemma 1, we have

$$\ell(\tilde{\theta}_{S,k}^{\text{ft}}) - \ell(\tilde{\theta}_S^*) \leq \frac{\|\tilde{\theta}^* - \tilde{\theta}_S^*\|_2^2}{2k\alpha}, \tag{32}$$

where we recall that $\tilde{\theta}^*$ denotes the pruned model before fine-tuning. Taking $k \to \infty$, we obtain $\ell(\tilde{\theta}_{S,k}^{\text{ft}}) - \ell(\tilde{\theta}_S^*) = 0$, and it follows that $\tilde{\theta}_{S,k}^{\text{ft}} = \tilde{\theta}_S^*$ because $\ell$ is convex. Therefore, as $k \to \infty$, we have

$$\mathbb{E}[|\mathcal{F}(\tilde{\theta}_{S,k}^{\text{ft}}) - \mathcal{F}(\theta_S^*)|] \leq \mathbb{E}[|\mathcal{F}(\tilde{\theta}_{S,k}^{\text{ft}}) - \mathcal{F}(\tilde{\theta}_S^*)|] + B \tag{33}$$

$$= \mathbb{E}[|\mathcal{F}(\tilde{\theta}_S^*) - \mathcal{F}(\tilde{\theta}_S^*)|] + B \tag{34}$$

$$= B \tag{35}$$

$\square$

Before proving Proposition 2, we provide the following lemma.

**Lemma 2.** *The exact Shapley values $\beta \in \mathbb{R}^n$ for the data contributors evaluated with $\{\mathcal{F}(\theta_S)\}_{S \in 2^{\mathcal{D}}}$ can be represented as $\beta = Av$, where $A \in \mathbb{R}^{n \times 2^n}$ is a matrix with rows indexed by the data contributors and columns indexed by all possible contributor subsets, and $v \in \mathbb{R}^{2^n}$ is a vector indexed by all possible contributor subsets. Specifically,*

$$A[i, S] = \begin{cases} \frac{1}{n} \binom{n-1}{|S|-1}^{-1}, & \text{if } C_i \in S \\ -\frac{1}{n} \binom{n-1}{|S|}^{-1}, & \text{otherwise} \end{cases} \tag{36}$$

*and $v[S] = \mathcal{F}(\theta_S)$.*

*Proof.* For each data contributor $i$, we have

$$\beta_i = \sum_{S \in 2^{\mathcal{D}}} A[i, S] \cdot v[S] \tag{37}$$

$$= \sum_{S:C_i \in S} \frac{1}{n} \binom{n-1}{|S|-1}^{-1} \mathcal{F}(\theta_S) - \sum_{S':C_i \notin S'} \frac{1}{n} \binom{n-1}{|S'|}^{-1} \mathcal{F}(\theta_{S'}) \tag{38}$$

$$= \sum_{S' \subseteq \mathcal{D} \backslash C_i} \frac{1}{n} \binom{n-1}{|S'|}^{-1} \mathcal{F}(\theta_{S' \cup C_i}) - \sum_{S' \subseteq \mathcal{D} \backslash C_i} \frac{1}{n} \binom{n-1}{|S'|}^{-1} \mathcal{F}(\theta_{S'}) \tag{39}$$

$$= \frac{1}{n} \sum_{S' \subseteq \mathcal{D} \backslash C_i} \binom{n-1}{|S'|}^{-1} (\mathcal{F}(\theta_{S' \cup C_i}) - \mathcal{F}(\theta_{S'})), \tag{40}$$

which is the Shapley value for contributor $i$. The third equality follows from the change of variable $S' = S \backslash C_i$ such that $|S'| = |S| - 1$. $\square$

Finally, we prove Proposition 2 as follows.

*Proof of Proposition 2.* Based on Lemma 2, $\tilde{\beta}_k^{\text{ft}} = A\tilde{v}_k^{\text{ft}}$, and $\beta^* = Av^*$, where $\tilde{v}_k^{\text{ft}}[S] = \mathcal{F}(\tilde{\theta}_{S,k}^{\text{ft}})$ and $v^*[S] = \mathcal{F}(\theta_S^*)$ for all $S \in 2^{\mathcal{D}}$. Hence we have

$$\mathbb{E}[\|\tilde{\beta}_k^{\text{ft}} - \beta^*\|_2] = \mathbb{E}[\|A\tilde{v}_k^{\text{ft}} - Av^*\|_2] \tag{41}$$

$$= \mathbb{E}\sqrt{\sum_{i=1}^{n} \left(A[i]^\top (\tilde{v}_k^{\text{ft}} - v^*)\right)^2} \tag{42}$$

$$\leq \mathbb{E}\sqrt{\sum_{i=1}^{n} \|A[i]\|_1^2 \cdot \|\tilde{v}_k^{\text{ft}} - v^*\|_\infty^2} \tag{43}$$

$$= \mathbb{E}\|\tilde{v}_k^{\text{ft}} - v^*\|_\infty \cdot \sqrt{\sum_{i=1}^{n} \|A[i]\|_1^2}, \tag{44}$$

where the inequality follows from Holder's inequality.

For each $i = 1, ..., n$, we have

$$\|A[i]\|_1 = \sum_{S \in 2^{\mathcal{D}}} |A[i, S]| \tag{45}$$

$$= \sum_{z=1}^{n} \sum_{S:C_i \in S, |S|=z} \frac{1}{n} \binom{n-1}{z-1}^{-1} + \sum_{z=0}^{n-1} \sum_{S:C_i \notin S, |S|=z} \frac{1}{n} \binom{n-1}{z}^{-1} \tag{46}$$

$$= \sum_{z=1}^{n} \frac{1}{n} \binom{n-1}{z-1}^{-1} \cdot \binom{n-1}{z-1} + \sum_{z=0}^{n-1} \frac{1}{n} \binom{n-1}{z}^{-1} \cdot \binom{n-1}{z} \tag{47}$$

$$= \frac{1}{n} \cdot n + \frac{1}{n} \cdot n = 2, \tag{48}$$

where the third equality follows from counting the number of subsets. Hence, we have

$$\mathbb{E}[\|\tilde{\beta}_k^{\text{ft}} - \beta^*\|_2] \leq 2\sqrt{n} \cdot \mathbb{E}\|\tilde{v}_k^{\text{ft}} - v^*\|_\infty \tag{49}$$

$$= 2\sqrt{n} \cdot \mathbb{E}\left[\max_{S \in 2^{\mathcal{D}}} |\mathcal{F}(\tilde{\theta}_{S,k}^{\text{ft}}) - \mathcal{F}(\theta_S^*)|\right] \tag{50}$$

$$\leq 2\sqrt{n} \cdot \left(\mathbb{E}[\max_{S \in 2^{\mathcal{D}}} |\mathcal{F}(\tilde{\theta}_{S,k}^{\text{ft}}) - \mathcal{F}(\tilde{\theta}_S^*)|] + \mathbb{E}[\max_{S \in 2^{\mathcal{D}}} |\mathcal{F}(\tilde{\theta}_S^*) - \mathcal{F}(\theta_S^*)|]\right) \tag{51}$$

$$= 2\sqrt{n} \cdot \mathbb{E}\left[\max_{S \in 2^{\mathcal{D}}} |\mathcal{F}(\tilde{\theta}_S^*) - \mathcal{F}(\theta_S^*)|\right], \tag{52}$$

as $k \to \infty$, where the last equality follows the same steps as in Proposition 1. Finally, with the assumption that $\mathbb{E}[\max_{S \in 2^{\mathcal{D}}} |\mathcal{F}(\tilde{\theta}_S^*) - \mathcal{F}(\theta_S^*)|] \leq C$, we have

$$\mathbb{E}[\|\tilde{\beta}_k^{\text{ft}} - \beta^*\|_2] \leq 2\sqrt{n}C. \tag{53}$$

$\square$

## B  DATASETS AND MODEL PROPERTIES

**CIFAR-20 (32 × 32).** The full CIFAR-100 dataset[8] contains 50,000 training samples across 100 classes, with each class labeled by a labeler (Krizhevsky et al., 2009). These labelers were compensated to ensure the removal of initially mislabeled images. For computational feasibility in the LDS evaluations, we follow a similar approach in Zheng et al. (2023) to create a smaller subset, CIFAR-20. This subset includes 10,000 training samples randomly selected from four superclasses in CIFAR-100: "large carnivores", "flowers", "household electrical devices", and "vehicles 1". For Inception Score calculation, we generate 10,240 samples and use Inception Score Pytorch [9].

**CelebA-HQ (256 × 256).** The original CelebA-HQ dataset (Karras et al., 2018) comprises of 30,000 face images with 6,217 unique identities. We downloaded a preprocessed dataset from `https://github.com/ndb796/LatentHSJA`, which contain 5,478 images from 307 identities, each with 15 or more images. For our experiment, we randomly sample 50 identities, totaling 895 images. Also, to match the input dimension required by the implementation of latent diffusion model, we resize the original images from a resolution of $1024 \times 1024$ to $256 \times 256$. To calculate the diversity score based on Equation (8), we first generate 1,024 samples from the model trained on the full dataset and extract image embeddings using the pretrained BLIP-VQA image encoder[10] (Li et al., 2022). These embeddings are then clustered into 20 distinct groups as the reference clusters using the Ward linkage criterion (Jh Jr, 1963), representing groups with demographic characteristics. Subsequently, for models trained on various subsets of celebrities, we generate 1,024 samples and assign them into these 20 reference clusters.

**ArtBench Post-Impressionism (256 × 256).** ArtBench-10 is a dataset for benchmarking artwork generation with 10 art styles, consisting of 5,000 training images per style (Liao et al., 2022). For our experiments, we consider the 5,000 training images corresponding to the art style "post-impressionism" from 258 artists. The aesthetic score predictor[11] is a linear model built on top of CLIP to predict the aesthetic quality of images as a proxy to how much people, on average, perceive an image as aesthetically pleasing.

## C  ADDITIONAL DETAILS ON TRAINING AND INFERENCE

**CIFAR-20 and CelebA-HQ.**  For CIFAR-20, we follow the original implementation of the unconditional DDPMs (Ho et al., 2020) where the model has 35.7M parameters. For CelebA-HQ, we follow the implementation of LDM (Rombach et al., 2022) with 274M parameters and a pre-trained VQ-VAE (Razavi et al., 2019). For both datasets, the maximum diffusion time step is set to $T = 1000$ during training, with a linear variance schedule for the forward diffusion process ranging from $\beta_1 = 10^{-4}$ to $\beta_T = 0.02$. For both CIFAR-20 and CelebA-HQ, we use the Adam optimizer (Loshchilov & Hutter, 2017) and apply random horizontal flipping as data augmentation. Both the DDPM and LDM are trained for 20,000 steps with a batch size of 64 and a learning rate of $10^{-4}$. During inference, images are generated using the 100-step DDIM solver (Song et al., 2020a).

**ArtBench (Post-Impressionism).**  A Stable Diffusion model[12] (Rombach et al., 2022) is fine-tuned using LoRA (Hu et al., 2021) with rank = 256, corresponding to 5.1M LoRA parameters. The prompt is set to *"a Post-Impressionist painting"* for each image. The LoRA parameters are trained using the AdamW optimizer (Loshchilov & Hutter, 2017) with a weight decay of $10^{-6}$, for 200 epochs and

---

[8] `https://www.cs.toronto.edu/~kriz/cifar.html`
[9] `https://github.com/sbarratt/inception-score-pytorch`
[10] `https://huggingface.co/Salesforce/blip-vqa-base`
[11] `https://github.com/LAION-AI/aesthetic-predictor`
[12] `https://huggingface.co/lambdalabs/miniSD-diffusers`

a batch size of 64. Cosine learning rate annealing is used, with 500 warm-up steps and an initial learning rate of $3 \times 10^{-4}$. At inference time, images are generated using the PNDM scheduler with 100 steps (Karras et al., 2022).

For training diffusion models, we utilize NVIDIA GPUs: RTX 6000, A40, and A100, equipped with 24GB, 48GB, and 80GB of memory, respectively. This study employs the Diffusers package version 0.24.0 [13] to train models across. Per-sample gradients are computed using the techniques outlined in the PyTorch package tutorial (version 2.2.1) [14]. Furthermore, we apply the TRAK package[15] to project gradients with a random projection matrix. All experiments are conducted on systems equipped with 64 CPU cores and the specified NVIDIA GPUs.

For CelebA-HQ, we implement two techniques to reduce GPU memory usage during training. First, we pre-compute the embeddings of VQ-VAE for all training samples, since the VQ-VAE is kept frozen during training. This approach allows us to avoid loading the VQ-VAE module into GPU memory during training. Second, we use the 8-bit Adam optimizer implemented in the bitsandbytes Python package (Dettmers et al., 2022).

# D    EMPIRICAL VERIFICATION OF PROPOSITION 1 AND COROLLARY 1

Proposition 1 suggests that increasing the number of fine-tuning steps can lead to a better approximation of the retrained model property $\mathcal{F}(\theta^*_{S_j})$ using the sparsified fine-tuned model property $\mathcal{F}(\tilde{\theta}^{\text{ft}}_{S_j,k})$. Here, we empirically measure the similarity between $\mathcal{F}(\theta^*_{S_j})$ and $\mathcal{F}(\tilde{\theta}^{\text{ft}}_{S_j,k})$, with 100 subsets $S_j$ sampled from the Shapley kernel and varying the number of sparsified fine-tuning steps $k$. As Proposition 1 provides an upper bound on the estimation error, the possibility of a constant shift should be considered[16]. Therefore, we focus on the Pearson correlation between $\{\mathcal{F}(\theta^*_{S_j})\}_{j=1}^{100}$ and $\{\mathcal{F}(\tilde{\theta}^{\text{ft}}_{S_j,k})\}_{j=1}^{100}$ as the similarity metric. As shown in Figure 5, the approximation generally becomes better with more fine-tuning steps.

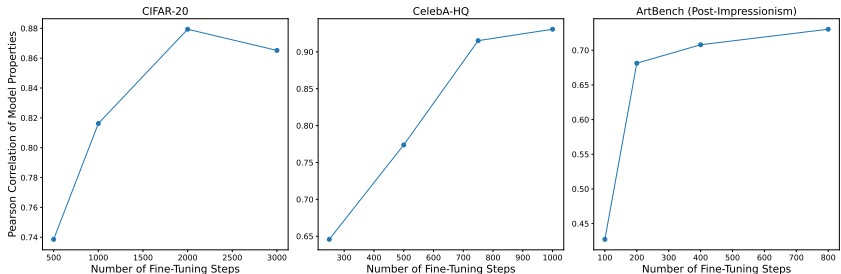

Figure 5: Pearson correlation between model behaviors evaluated with retraining vs. sparsified fine-tuning, with varying number of fine-tuning steps. Models are retrained or fine-tuned on 100 contributor subsets sampled from the Shapley kernel.

Proposition 2 suggests that increasing the number of fine-tuning steps can also lead to a better approximation of the retrained Shapley values (denoted as $\beta^*$) using Shapley values estimated through sparsified fine-tuning (denoted as $\tilde{\beta}^{\text{ft}}_k$). Here, we compare the correlation between $\beta^*$ and $\tilde{\beta}^{\text{ft}}_k$, both estimated with 500 contributor subsets, with varying number of fine-tuning steps $k$. As shown in Figure 6, the approximation tends to become better with more fine-tuning steps.

Finally, we note there is a trade-off between better approximation and computational efficiency. As shown in Figure 7, average runtime increases linearly with more fine-tuning steps.

---

[13]https://pypi.org/project/diffusers/

[14]https://pytorch.org/

[15]https://trak.csail.mit.edu/quickstart

[16]For example, $\mathbb{E}[|\mathcal{F}(\tilde{\theta}^{\text{ft}}_{S_j,k}) - \mathcal{F}(\theta^*_{S_j})|] \leq B$ still holds when $\mathcal{F}(\tilde{\theta}^{\text{ft}}_{S_j,k}) = \mathcal{F}(\theta^*_{S_j}) + B'$ for some $-B \leq B' \leq B$

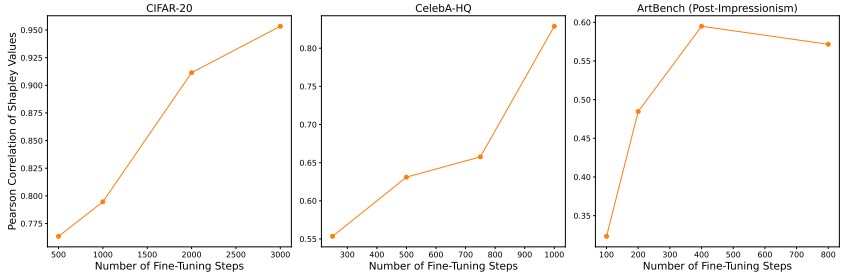

Figure 6: Pearson correlation between the Shapley values estimated with retraining vs. sparsified fine-tuning, with varying number of fine-tuning steps. Shapley values are estimated with 500 contributor subsets sampled from the Shapley kernel.

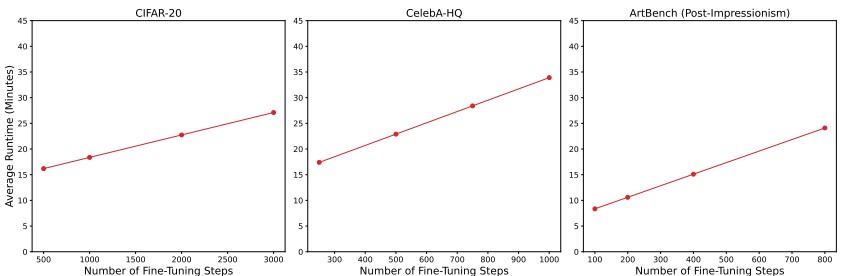

Figure 7: Average runtime of sparsified fine-tuning for the Shapley kernel.

## E    DATA ATTRIBUTION METHODS FOR DIFFUSION MODELS

Here we provide definitions and implementation details of the baselines used in Section 4. For TRAK-based approaches, we follow the implementation details in Zheng et al. (2023)[17].

**Raw pixel similarity.** This is a simple baseline that uses each raw image as the representation and then computes the cosine similarity between the generated sample of interest and each training sample as the attribution score.

**Embeddings distance.** This method computes the $l_2$ distances between training samples encoded using BLIP-VQA[18] (Li et al., 2022), and the average embedding of each reference cluster as the attribution score.

**CLIP similarity.** This method utilizes CLIP[19] (Radford et al., 2021) to encode each sample into an embedding and then compute the cosine similarity between the generated sample of interest and each training sample as the attribution score.

**Leave-one-out.** Leave-one-out (LOO) evaluates the change in model property due to the removal of a single data contributor $i$ from the training set (Koh & Liang, 2017). Formally, it is defined as:

$$\tau_{\text{LOO}}(\mathcal{F}, \mathcal{D})_i = \mathcal{F}(\theta^*) - \mathcal{F}(\theta^*_{\mathcal{D} \setminus \mathcal{C}_i}), \tag{54}$$

where $\mathcal{F}(\theta^*)$ denotes the global property for the model trained with the full dataset, and $\mathcal{F}(\theta^*_{\mathcal{D} \setminus \mathcal{C}_i})$ represents the global property of the mode trained with the data provided by the $i$th contributor.

**Gradient similarity.** This is a gradient-based influence estimator (Charpiat et al., 2019), which computes the cosine similarity using the gradient representations of the generated sample of interest, $\tilde{\mathbf{x}}$ and each training sample $\mathbf{x}^{(j)}$ as the attribution score:

$$\frac{\mathcal{P}^{\mathsf{T}} \nabla_\theta \mathcal{L}_{\text{Simple}}(\mathbf{x}^{(j)}; \theta^*) \cdot \mathcal{P}^{\mathsf{T}} \nabla_\theta \mathcal{L}_{\text{Simple}}(\tilde{\mathbf{x}}; \theta^*)}{\|\mathcal{P}^{\mathsf{T}} \nabla_\theta \mathcal{L}_{\text{Simple}}(\mathbf{x}^{(j)}; \theta^*)^{\mathsf{T}}\| \|\mathcal{P}^{\mathsf{T}} \nabla_\theta \mathcal{L}_{\text{Simple}}(\tilde{\mathbf{x}}; \theta^*)\|} \tag{55}$$

---

[17]https://github.com/sail-sg/D-TRAK/

[18]https://huggingface.co/Salesforce/blip-vqa-base

[19]https://github.com/openai/CLIP

**TRAK.** Park et al. (2023) propose an approach that aims to enhance the efficiency and scalability of data attribution for classifiers. Zheng et al. (2023) adapt TRAK for attributing images generated by diffusion models, defined as follows:

$$\frac{1}{S} \sum_{s=1}^{S} \Phi_s \left( \Phi_s^\top \Phi_s + \lambda I \right)^{-1} \mathcal{P}_s^\top \nabla_\theta \mathcal{L}_{\text{Simple}}(\tilde{\mathbf{x}}; \theta_s^*), \text{ and} \tag{56}$$

$$\Phi_s = \left[ \phi_s(\mathbf{x}^{(1)}), \dots, \phi_s(\mathbf{x}^{(N)}) \right]^\top, \text{ where } \phi_s(\mathbf{x}) = \mathcal{P}_s^\top \nabla_\theta \mathcal{L}_{\text{Simple}}(\mathbf{x}; \theta_s^*), \tag{57}$$

where $\mathcal{L}_{\text{Simple}}$ is the diffusion loss used during training, $\mathcal{P}_s$ is a random projection matrix, and $\lambda I$ serves for numerical stability and regularization.

**D-TRAK.** Zheng et al. (2023) find that, compared to the theoretically motivated setting where the loss function is defined as $\mathcal{L} = \mathcal{L}_{\text{Simple}}$, using alternative functions—such as the squared loss $\mathcal{L}_{\text{Square}}$—to replace both the loss function and local model property result in improved performance:

$$\frac{1}{S} \sum_{s=1}^{S} \Phi_s \left( \Phi_s^\top \Phi_s + \lambda I \right)^{-1} \mathcal{P}_s^\top \nabla_\theta \mathcal{L}_{\text{Square}}(\tilde{\mathbf{x}}; \theta_s^*), \text{ and} \tag{58}$$

$$\Phi_s = \left[ \phi_s(\mathbf{x}^{(1)}), \dots, \phi_s(\mathbf{x}^{(N)}) \right]^\top, \text{ where } \phi_s(\mathbf{x}) = \mathcal{P}_s^\top \nabla_\theta \mathcal{L}_{\text{Square}}(\mathbf{x}; \theta_s^*). \tag{59}$$

**Journey-TRAK.** Georgiev et al. (2023) focus on attributing noisy images $\mathbf{x}_t$ during the denoising process. It attributes training data not only for the final generated sample but also for the intermediate noisy samples throughout the denoising process. At each time step, the reconstruction loss is treated as a local model property. Following (Zheng et al., 2023), we compute attribution scores by averaging over the generation time steps, as shown below:

$$\frac{1}{T'} \frac{1}{S} \sum_{t=1}^{T'} \sum_{s=1}^{S} \Phi_s \left( \Phi_s^\top \Phi_s + \lambda I \right)^{-1} \mathcal{P}_s^\top \nabla_\theta \mathcal{L}_{\text{Simple}}^t(\tilde{\mathbf{x}}_t; \theta_s^*), \text{ and} \tag{60}$$

$$\Phi_s = \left[ \phi_s(\mathbf{x}^{(1)}), \dots, \phi_s(\mathbf{x}^{(N)}) \right]^\top, \text{ where } \phi_s(\mathbf{x}) = \mathcal{P}_s^\top \nabla_\theta \mathcal{L}_{\text{Simple}}(\mathbf{x}; \theta_s^*). \tag{61}$$

Here, $\mathcal{L}_{\text{Simple}}^t$ is the diffusion loss restricted to the time step $t$ (hence without taking the expectation over the time step).

In practice, we follow the main implementation by Zheng et al. (2023) that considers TRAK, D-TRAK, and Journey-TRAK as retraining-free methods. That is, $S = 1$, and $\theta_1^* = \theta^*$ is the original diffusion model trained on the entire dataset.

**Relative Influence.** Proposed by Barshan et al. (2020), the $\theta$-relative influence functions estimator normalizes the influence functions estimator of Koh & Liang (2017) by the HVP magnitude. Following Zheng et al. (2023), we combine this estimator with TRAK dimension reudction. The attribution for each training sample $\mathbf{x}^{(j)}$ is as follows:

$$\frac{\mathcal{P}^\top \nabla_\theta \mathcal{L}_{\text{Simple}}(\tilde{\mathbf{x}}; \theta^*) \cdot \left( \Phi_{\text{TRAK}}^\top \cdot \Phi_{\text{TRAK}} + \lambda I \right)^{-1} \cdot \mathcal{P}^\top \nabla_\theta \mathcal{L}_{\text{Simple}}(\mathbf{x}^{(j)}; \theta^*)}{\left\| \left( \Phi_{\text{TRAK}}^\top \cdot \Phi_{\text{TRAK}} + \lambda I \right)^{-1} \cdot \mathcal{P}^\top \nabla_\theta \mathcal{L}_{\text{Simple}}(\mathbf{x}^{(j)}; \theta^*) \right\|}, \tag{62}$$

and

$$\Phi_{\text{TRAK}} = \left[ \phi(\mathbf{x}^{(1)}), \dots, \phi(\mathbf{x}^{(N)}) \right]^\top, \text{ where } \phi(\mathbf{x}) = \mathcal{P}^\top \nabla_\theta \mathcal{L}_{\text{Simple}}(\mathbf{x}; \theta^*). \tag{63}$$

**Renormalized Influence.** Introduced by Hammoudeh & Lowd (2022), this method renormalizes the influence functions by the magnitude of the training sample's gradients. Similar to relative influence, we made an adaptation following Zheng et al. (2023). The attribution for each training sample $\mathbf{x}^{(j)}$ is as follows:

$$\frac{\mathcal{P}^\top \nabla_\theta \mathcal{L}_{\text{Simple}}(\tilde{\mathbf{x}}; \theta^*) \cdot \left( \Phi_{\text{TRAK}}^\top \cdot \Phi_{\text{TRAK}} + \lambda I \right)^{-1} \cdot \mathcal{P}^\top \nabla_\theta \mathcal{L}_{\text{Simple}}(\mathbf{x}^{(j)}; \theta^*)}{\left\| \mathcal{P}^\top \nabla_\theta \mathcal{L}_{\text{Simple}}(\mathbf{x}^{(j)}; \theta^*) \right\|}, \tag{64}$$

and

$$\Phi_{\text{TRAK}} = \left[ \phi(\mathbf{x}^{(1)}), \ldots, \phi(\mathbf{x}^{(N)}) \right]^{\top}, \text{ where } \phi(\mathbf{x}) = \mathcal{P}^{\top} \nabla_{\theta} \mathcal{L}_{\text{Simple}}(\mathbf{x}; \theta^*). \tag{65}$$

For TRAK-based methods and gradient similarity, we compute gradients using the final model checkpoint, meaning that $S = 1$ in Equation (56). We select 100 time steps evenly spaced within the interval $[1, T]$ for computing the diffusion loss. At each time step, we introduce one instance of noise. The projection dimensions are set to $k = 4096$ for CIFAR-20 and $k = 32768$ for CelebA-HQ and ArtBench (Post-Impressionism). For D-TRAK, we use the best-performing output function $\mathcal{L}_{\text{Square}}$ and choose $\lambda = 5e^{-1}$ as in Zheng et al. (2023).

**TracIn.** Pruthi et al. (2020) propose to attribute the training sample's influence based on first-order approximation with saved checkpoints during the training process. However, this is also a major limitation because sometimes it is impossible to obtain checkpoints for models such as Stable Diffusion (Rombach et al., 2022).

# F    ADDITIONAL EXPERIMENT RESULTS

In this section, we present additional experiment results, including 1) LDS results across different datamodel alpha and computational budgets, 2) generated images after removing top contributing groups, 3) the performance of alternative unlearning approaches in CIFAR-20, and 4) analysis for top contributors in each dataset.

Table 2: Average runtime for data subset in minutes (training + inference) for Shapley values estimated with retraining, fine-tuning (FT), sparsified FT, and LoRA fine-tuning.

| Method | CIFAR-20 | CelebA-HQ | ArtBench (Post-Impressionism) |
|---|---|---|---|
| Retrain | 77.4 + 20.8 | 213.4 + 26.5 | 190.6 + 6.4 |
| FT | 6.06 + 18.9 | 17.5 + 27.0 | 4.4 + 6.4 |
| Sparsified FT | **4.37 + 14.0** | **11.0 + 11.9** | **4.5 + 6.1** |
| LoRA | 4.21 + 19.0 | 5.5 + 26.7 | - |

First, we provide the training cost incurred for each method. Utilizing 8 RTX-6k GPUs, the total runtime cost for sparsified-FT Shapley is under one day for each dataset. In contrast, the corresponding runtime costs for retraining-based Shapley are 4.3 days, 10.4 days, and 17.1 days. This highlights that sparsified-FT Shapley is more computationally feasible than retraining-based Shapley.

## F.1    CIFAR-20

Table 3: LDS (%) results with $\alpha = 0.25, 0.5, 0.75$ on CIFAR-20, with the Inception Score of 10,240 generated images as the global model property. Means and 95% confidence intervals across three random initializations are reported.

| Method | $\alpha = 0.25$ | $\alpha = 0.5$ | $\alpha = 0.75$ |
|---|---|---|---|
| Pixel similarity (average) | -10.88 $\pm$ 9.61 | -11.81 $\pm$ 4.56 | -20.82 $\pm$ 5.57 |
| Pixel similarity (max) | -17.42 $\pm$ 0.83 | -31.80 $\pm$ 2.90 | -23.88 $\pm$ 1.28 |
| CLIP similarity (average) | -0.34 $\pm$ 4.65 | -21.27 $\pm$ 1.37 | -13.13 $\pm$ 1.96 |
| CLIP similarity (max) | -1.19 $\pm$ 15.19 | 11.31 $\pm$ 0.37 | 16.05 $\pm$ 7.30 |
| Gradient similarity (average) | 6.48 $\pm$ 10.98 | 5.79 $\pm$ 3.67 | -10.75 $\pm$ 7.06 |
| Gradient similarity (max) | 7.22 $\pm$ 3.88 | -0.89 $\pm$ 3.17 | -4.82 $\pm$ 4.71 |
| Relative IF | 1.14 $\pm$ 13.51 | 5.23 $\pm$ 5.50 | 5.21 $\pm$ 3.78 |
| Renormalized IF | 9.09 $\pm$ 13.40 | 11.39 $\pm$ 6.79 | 7.99 $\pm$ 4.02 |
| TRAK | 3.62 $\pm$ 14.02 | 7.94 $\pm$ 5.67 | 6.59 $\pm$ 3.88 |
| Journey-TRAK | -30.09 $\pm$ 3.85 | -42.92 $\pm$ 2.15 | -40.43 $\pm$ 3.46 |
| D-TRAK | 21.02 $\pm$ 6.76 | 10.90 $\pm$ 1.21 | 21.90 $\pm$ 5.02 |
| LOO | 17.01 $\pm$ 5.29 | 30.66 $\pm$ 6.11 | 13.64 $\pm$4.99 |
| Sparsified-FT Shapley (**Ours**) | **51.24 $\pm$ 3.39** | **61.48 $\pm$ 2.27** | **59.15 $\pm$ 4.24** |

Table 4: LDS (%) results on Shapley value with different number of subset $S$.

| $\alpha$ | Method | $S = 100$ | $S = 200$ | $S = 300$ | $S = 400$ | $S = 500$ |
|---|---|---|---|---|---|---|
| 0.25 | Retraining | 43.88 (3.97) | 49.95 (3.20) | 57.56 (2.52) | 62.68 (1.85) | 65.90 (1.52) |
| | Sparsified-FT | 8.46 (13.41) | 23.92 (6.27) | 33.88 (8.33) | 54.88 (7.44) | 51.24 (3.39) |
| | FT | 13.57 (7.09) | 40.01 (8.30) | 28.94 (8.59) | 24.58 (6.07) | 20.57 (3.02) |
| | LoRA | 12.00 (0.65) | 29.61 (1.37) | 54.26 (3.38) | 51.55 (6.40) | 61.60 (7.44) |
| 0.5 | Retraining | 48.84 (2.00) | 39.55 (3.17) | 57.24 (2.69) | 66.81 (2.99) | 70.58 (2.05) |
| | Sparsified-FT | -5.41 (4.78) | 38.00 (3.06) | 47.74 (1.49) | 63.62 (2.30) | 61.48 (2.27) |
| | FT | 13.57 (7.09) | 40.01 (8.30) | 28.94 (8.59) | 24.58 (6.07) | 39.60 (3.03) |
| | LoRA | 29.90 (2.96) | 41.41 (1.28) | 41.27 (0.59) | 34.20 (0.53) | 30.86 (1.07) |
| 0.75 | Retraining | 51.39 (3.21) | 45.85 (3.99) | 61.91 (4.62) | 70.59 (3.13) | 72.07 (4.83) |
| | Sparsified-FT | -11.44 (2.26) | 33.98 (0.98) | 48.16 (5.18) | 59.65 (5.34) | 59.15 (4.24) |
| | FT | 33.90 (5.88) | 26.09 (8.55) | 45.49 (6.83) | 46.92 (5.96) | 38.51 (4.62) |
| | LoRA | 24.55 (2.16) | 31.07 (7.83) | 32.67 (7.94) | 35.01 (7.05) | 35.16 (8.14) |

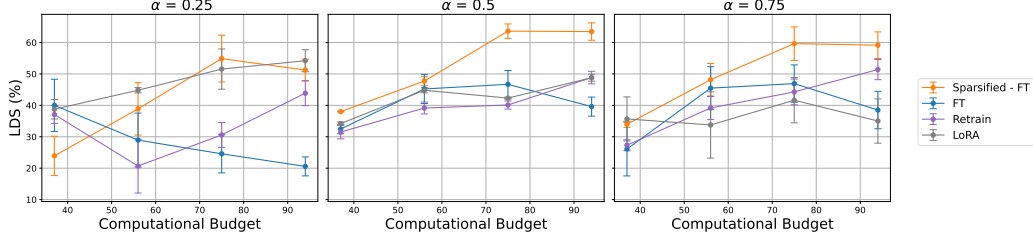

Figure 8: LDS (%) results on CIFAR-20 for Shapley values estimated with sparsified fine-tuning (FT), finet-tuning (FT), LoRA fine-tuning, and retraining under the same computational budgets (1 unit = runtime to retrain and run inference on a full model).

Table 5: LDS (%) results for retraining-based attribution across Shapley, Leave-One-Out (LOO), and Banzhaf distributions with $\alpha = 0.25, 0.5, 0.75$ on CIFAR-20. The global model behavior is evaluated using the Inception Score of 10,240 generated images. Means and 95% confidence intervals across three random initializations are reported. For sparsified fine-tuning (sFT) and fine-tuning (FT), the number of fine-tuning steps is set to 1000.

| Method | $\alpha = 0.25$ | $\alpha = 0.5$ | $\alpha = 0.75$ |
|---|---|---|---|
| LOO (retraining) | $17.01 \pm 5.29$ | $30.66 \pm 6.11$ | $13.64 \pm 4.99$ |
| Banzhaf (retraining) | $10.59 \pm 8.64$ | $37.11 \pm 2.33$ | $42.78 \pm 1.85$ |
| Shapley (retraining) | $\mathbf{65.90 \pm 1.52}$ | $\mathbf{70.58 \pm 2.05}$ | $\mathbf{72.07 \pm 4.83}$ |
| LOO (FT) | $-55.00 \pm 11.76$ | $-66.06 \pm 2.28$ | $-54.58 \pm 7.02$ |
| Banzhaf (FT) | $-11.20 \pm 6.42$ | $9.09 \pm 2.67$ | $16.79 \pm 1.12$ |
| Shapley (FT) | $\mathbf{20.57 \pm 3.02}$ | $\mathbf{39.60 \pm 3.03}$ | $\mathbf{38.51 \pm 4.62}$ |
| LOO (sFT) | $29.45 \pm 5.96$ | $27.43 \pm 4.20$ | $19.58 \pm 0.35$ |
| Banzhaf (sFT) | $5.44 \pm 8.59$ | $22.55 \pm 5.07$ | $31.44 \pm 0.27$ |
| Shapley (sFT) | $\mathbf{51.24 \pm 3.39}$ | $\mathbf{61.48 \pm 2.27}$ | $\mathbf{59.15 \pm 4.24}$ |

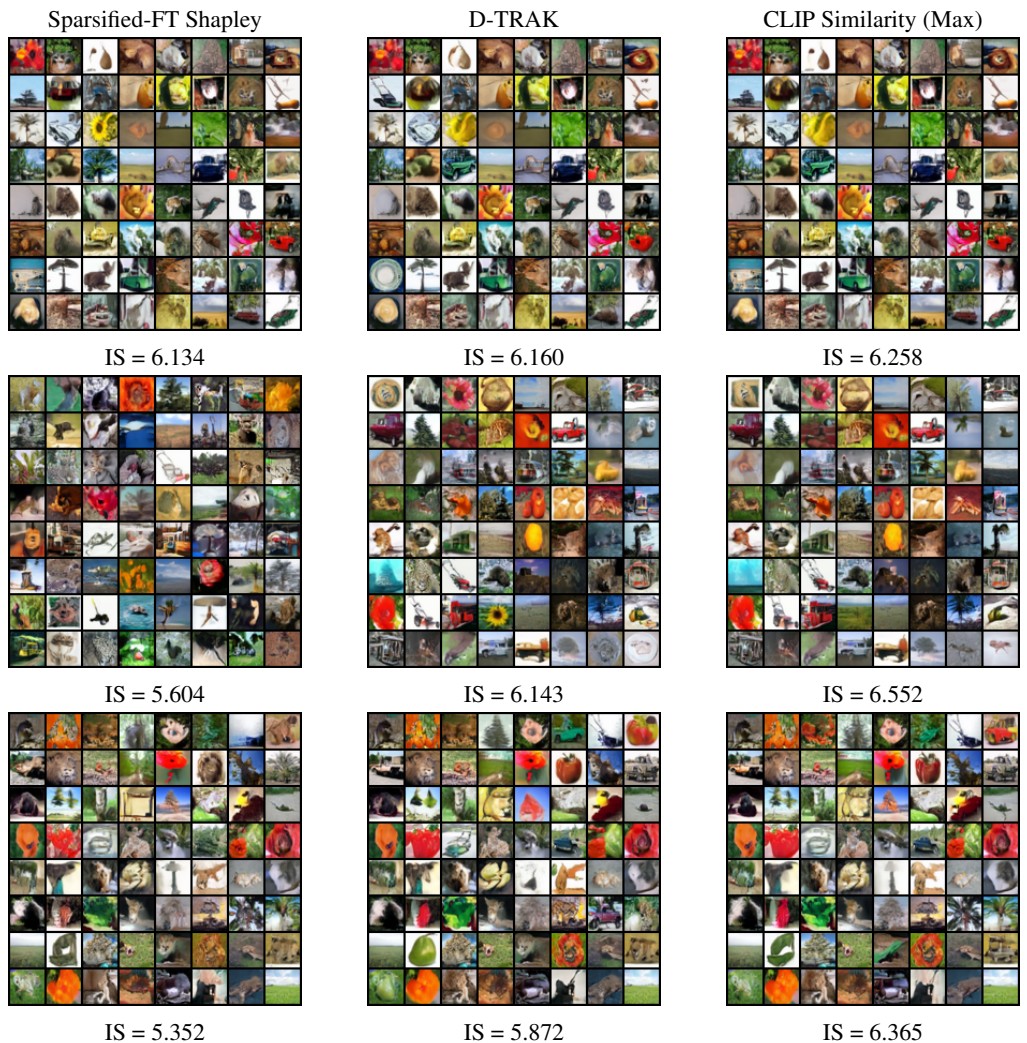

Figure 9: Inception scores and example images generated with the same initial noises for DDPMs trained without the 10%, 20%, and 40% (top to bottom) most important CIFAR-20 classes based on sparsified-FT Shapley, D-TRAK, and CLIP similarity (max).

### F.2 CELEBA-HQ

Table 6: LDS (%) results with $\alpha = 0.25, 0.5, 0.75$ on CelebA-HQ, with the cluster entropy of 1,024 generated images as the global model property. Means and 95% confidence intervals across three random initializations are reported.

| Method | $\alpha = 0.25$ | $\alpha = 0.5$ | $\alpha = 0.75$ |
|---|---|---|---|
| Pixel similarity (average) | $-1.52 \pm 3.14$ | $-8.91 \pm 0.93$ | $-4.19 \pm 2.25$ |
| Pixel similarity (max) | $9.53 \pm 6.35$ | $21.70 \pm 2.05$ | $24.76 \pm 0.96$ |
| Embedding dist. (average) | $3.59 \pm 6.47$ | $13.83 \pm 1.12$ | $10.89 \pm 2.04$ |
| Embedding dist. (max) | $9.17 \pm 2.12$ | $7.32 \pm 3.16$ | $22.25 \pm 1.87$ |
| CLIP similarity (average) | $-8.29 \pm 4.38$ | $-32.23 \pm 0.87$ | $-22.29 \pm 2.56$ |
| CLIP similarity (max) | $-23.62 \pm 3.51$ | $-0.93 \pm 3.83$ | $-8.86 \pm 2.16$ |
| Gradient similarity (average) | $-19.40 \pm 3.81$ | $-18.32 \pm 0.65$ | $-21.29 \pm 1.76$ |
| Gradient similarity (max) | $-15.98 \pm 2.72$ | $-12.90 \pm 1.60$ | $-25.41 \pm 2.42$ |
| Relative IF | $-5.14 \pm 2.30$ | $-1.07 \pm 0.68$ | $-5.50 \pm 1.13$ |
| Renormalized IF | $-5.27 \pm 6.15$ | $10.17 \pm 0.57$ | $0.39 \pm 1.20$ |
| TRAK | $-6.59 \pm 5.04$ | $3.22 \pm 0.75$ | $-3.65 \pm 1.24$ |
| Journey-TRAK | $-16.82 \pm 3.25$ | $-2.88 \pm 4.02$ | $-12.66 \pm 2.27$ |
| D-TRAK | $-30.77 \pm 4.26$ | $-27.23 \pm 2.80$ | $-23.67 \pm 1.71$ |
| LOO | $11.9 \pm 8.32$ | $-1.22 \pm 6.34$ | $-8.47 \pm -7.93$ |
| Sparsified-FT Shapley **(Ours)** | $10.05 \pm 5.33$ | $\mathbf{26.34 \pm 3.34}$ | $24.08 \pm 1.52$ |

Table 7: LDS (%) results on Shapley value with different number of subsets $S$

| $\alpha$ | Method | $S = 100$ | $S = 200$ | $S = 300$ | $S = 400$ | $S = 500$ |
|---|---|---|---|---|---|---|
| 0.25 | Retraining | 3.54 (0.59) | 8.91 (2.61) | 16.77 (4.75) | 15.58 (6.23) | 31.52 (5.56) |
| | Sparsified - FT | 9.95 (5.77) | 15.93 (6.29) | 9.90 (8.17) | 16.12 (6.40) | 10.05 (5.33) |
| | FT | -10.52 (4.26) | 2.77 (5.79) | 12.29 (4.50) | 9.86 (4.84) | 13.91 (3.91) |
| | LoRA | 3.07 (1.17) | 19.03 (9.47) | 26.20 (9.73) | 27.17 (7.72) | 31.09 (7.72) |
| 0.5 | Retraining | 19.01 (1.60) | 28.43 (2.71) | 35.06 (3.48) | 23.48 (2.95) | 28.58 (3.91) |
| | Sparsified - FT | 20.73 (2.89) | 26.83 (3.96) | 12.71 (2.06) | 27.23 (2.94) | 26.34 (3.42) |
| | FT | 1.82 (3.36) | 9.25 (2.59) | 12.17 (2.83) | 8.63 (0.52) | 13.77 (1.42) |
| | LoRA | 10.66 (3.56) | 20.12 (2.72) | 13.99 (1.13) | 21.12 (0.85) | 22.37 (0.82) |
| 0.75 | Retraining | 21.14 (0.93) | 26.29 (1.37) | 31.87 (2.85) | 31.60 (2.23) | 32.26 (2.08) |
| | Sparsified - FT | 16.05 (0.31) | 29.35 (1.43) | 9.49 (2.03) | 17.70 (1.30) | 24.08 (1.52) |
| | FT | -11.02 (1.51) | -12.07 (4.30) | 1.93 (4.82) | 0.96 (4.16) | 7.37 (3.79) |
| | LoRA | 6.05 (1.17) | 9.66 (1.43) | -1.92 (0.77) | 9.00 (2.18) | 14.60 (2.33) |

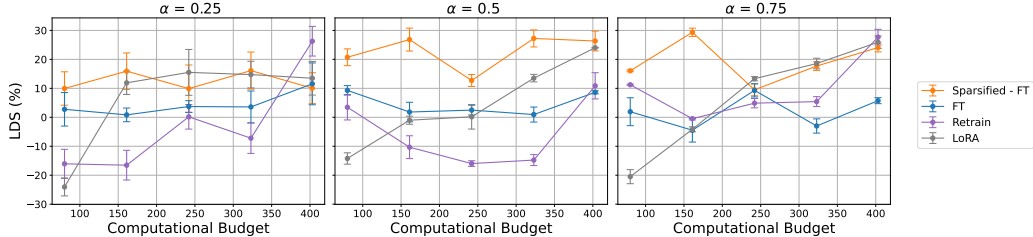

Figure 10: LDS (%) results on CelebA-HQ for Shapley values estimated with sparsified fine-tuning (FT), fine-tuning (FT), and retraining under the same computational budgets (1 unit = runtime to retrain and run inference on a full model).

Table 8: LDS (%) results for retraining-based attribution across Shapley, Leave-One-Out (LOO), and Banzhaf distributions with $\alpha = 0.25, 0.5, 0.75$. The global model behavior is calculated with the cluster entropy of 1,024 generated images. Means and 95% confidence intervals across three random initializations are reported. For sparsified fine-tuning (sFT) and fine-tuning (FT), the number of training steps is set to 500.

| Method | $\alpha = 0.25$ | $\alpha = 0.5$ | $\alpha = 0.75$ |
|---|---|---|---|
| LOO (retraining) | $11.90 \pm 8.32$ | $-1.22 \pm 6.34$ | $-8.20 \pm 0.27$ |
| Banzhaf (retraining) | $7.87 \pm 1.31$ | $6.79 \pm 1.27$ | $12.54 \pm 2.60$ |
| Shapley (retraining) | $\mathbf{31.52 \pm 5.56}$ | $\mathbf{28.58 \pm 3.91}$ | $\mathbf{32.26 \pm 2.08}$ |
| LOO (FT) | $-2.70 \pm 1.84$ | $-10.15 \pm 1.12$ | $-10.09 \pm 1.70$ |
| Banzhaf (FT) | $1.50 \pm 7.36$ | $-2.78 \pm 0.69$ | $3.23 \pm 1.91$ |
| Shapley (FT) | $\mathbf{13.91 \pm 3.91}$ | $\mathbf{13.77 \pm 1.42}$ | $\mathbf{7.37 \pm 3.79}$ |
| LOO (sFT) | $0.93 \pm 2.83$ | $-9.26 \pm 0.39$ | $-18.80 \pm 1.28$ |
| Banzhaf (sFT) | $7.87 \pm 1.31$ | $6.79 \pm 1.27$ | $12.54 \pm 2.60$ |
| Shapley (sFT) | $\mathbf{10.05 \pm 5.33}$ | $\mathbf{26.34 \pm 3.34}$ | $\mathbf{24.08 \pm 1.52}$ |

### F.3 ARTBENCH (POST-IMPRESSIONISM)

Table 9: LDS (%) results with $\alpha = 0.25, 0.5, 0.75$ on ArtBench (Post-Impressionism), with the 90th percentil of aesthetic scores for 50 generated images as the global model property. Means and 95% confidence intervals across three random initializations are reported.

| Method | $\alpha = 0.25$ | $\alpha = 0.5$ | $\alpha = 0.75$ |
|---|---|---|---|
| Pixel similarity (average) | 16.57 (3.16) | 11.24 (0.63) | -1.19 (5.71) |
| Pixel similarity (max) | 16.62 (2.95) | 14.61 (2.72) | 3.43 (10.57) |
| CLIP similarity (average) | -6.03 (1.61) | -6.96 (4.08) | -9.27 (2.74) |
| CLIP similarity (max) | 8.94 (0.43) | -1.75 (4.07) | -1.70 (9.89) |
| Grad similarity (average) | 4.14 (5.14) | 0.25 (1.18) | -7.81 (0.89) |
| Grad similarity (max) | 22.10 (9.02) | 10.48 (3.11) | 1.32 (4.19) |
| Aesthetic score (average) | 24.81 (3.00) | 24.85 (2.30) | 13.21 (7.82) |
| Aesthetic score (max) | 31.73 (3.04) | 21.36 (3.70) | 7.45 (11.66) |
| Relative IF | 3.30 (4.98) | -5.02 (1.77) | -3.77 (12.05) |
| Renormalized IF | -1.71 (4.37) | -11.41 (0.93) | -8.96 (12.86) |
| TRAK | -0.85 (4.74) | -8.18 (1.30) | -6.78 (13.37) |
| Journey-TRAK | -14.10 (5.09) | -11.41 (4.22) | -6.83 (2.80) |
| D-TRAK | 19.37 (3.29) | 11.30 (3.47) | 17.72 (6.87) |
| Sparsified-FT Shapley **(Ours)** | **52.83 (3.58)** | **61.44 (2.04)** | **32.24 (10.93)** |

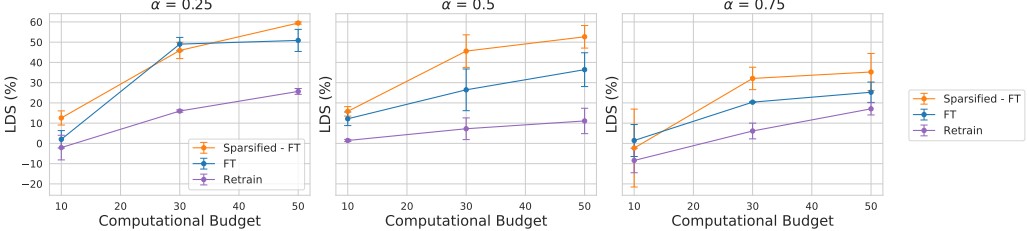

Figure 11: LDS (%) results with $\alpha = 0.25, 0.5, 0.75$ on ArtBench (Post-Impressionism) for Shapley values estimated with sparsified fine-tuning (FT), fine-tuning (FT), and retraining under the same computational budgets (1 unit = runtime to retrain and run inference on a full model).

Table 10: Aesthetic Score (90th percentile of 50 generated images) as the global model behavior. 100 datamodel subsets with alpha = 0.25, 0.5, 0.75 are used for evaluation. Means and 95% confidence intervals across three random initializations are reported. All Shapley values are estimated with 1,000 subsets.

| $\alpha$ | **Retrain** | **Sparsified - FT** | **FT** |
|---|---|---|---|
| 0.25 | 53.25 (6.75) | 51.15 (7.54) | 52.83 (3.58) |
| 0.50 | 52.30 (2.32) | 51.61 (3.21) | 61.44 (2.04) |
| 0.75 | 23.78 (14.07) | 33.77 (6.65) | 32.24 (10.93) |

Table 11: LDS (%) results with $\alpha = 0.25, 0.5, 0.75$ on ArtBench (Post-Impressionism), with the 90th percentile of aesthetic score for 50 generated images as the global model property. Means and 95% confidence intervals across three random initializations are reported. For sparsified fine-tuning (sFT) and fine-tuning (FT), the number of fine-tuning steps is set to 200. For Banzhaf and Shapley values, 1000 sbusets are used for estimation.

| Method | $\alpha = 0.25$ | $\alpha = 0.5$ | $\alpha = 0.75$ |
|---|---|---|---|
| LOO (retraining) | $2.69 \pm 4.57$ | $3.74 \pm 8.00$ | $-1.13 \pm 15.78$ |
| Banzhaf (retraining) | $4.70 \pm 7.28$ | $16.19 \pm 1.98$ | $-10.16 \pm 9.49$ |
| Shapley (retraining) | $\mathbf{53.25 \pm 6.75}$ | $\mathbf{52.30 \pm 2.32}$ | $\mathbf{23.78 \pm 14.07}$ |
| LOO (FT) | $-14.13 \pm 5.36$ | $-0.84 \pm 1.44$ | $6.62 \pm 6.72$ |
| Banzhaf (FT) | $3.55 \pm 6.99$ | $15.30 \pm 2.09$ | $-10.79 \pm 9.40$ |
| Shapley (FT) | $\mathbf{52.83 \pm 3.58}$ | $\mathbf{61.44 \pm 2.04}$ | $\mathbf{32.24 \pm 10.93}$ |
| LOO (sFT) | $2.94 \pm 2.64$ | $2.84 \pm 7.33$ | $1.00 \pm 4.45$ |
| Banzhaf (sFT) | $3.81 \pm 7.16$ | $15.51 \pm 2.15$ | $-10.43 \pm 9.16$ |
| Shapley (sFT) | $\mathbf{51.15 \pm 7.54}$ | $\mathbf{51.61 \pm 3.21}$ | $\mathbf{33.77 \pm 6.65}$ |

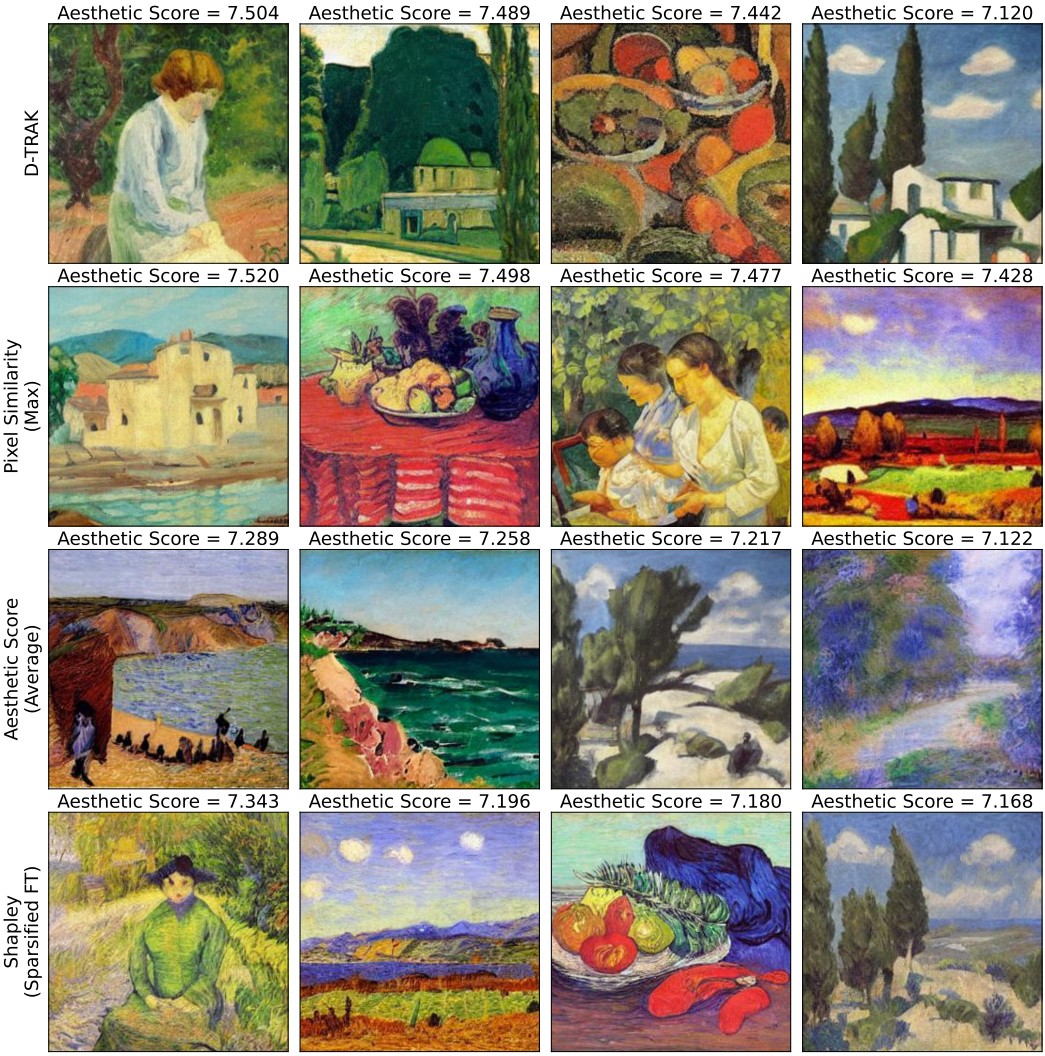

Figure 12: Two generated images above and two generated images below the 90th percentile of aesthetic scores, for Stable Diffusion models LoRA-finetuned without the top 40% most important artists based on D-TRAK, pixel similarity (max), training image aesthetic score (average), and sparsified-FT Shapley.

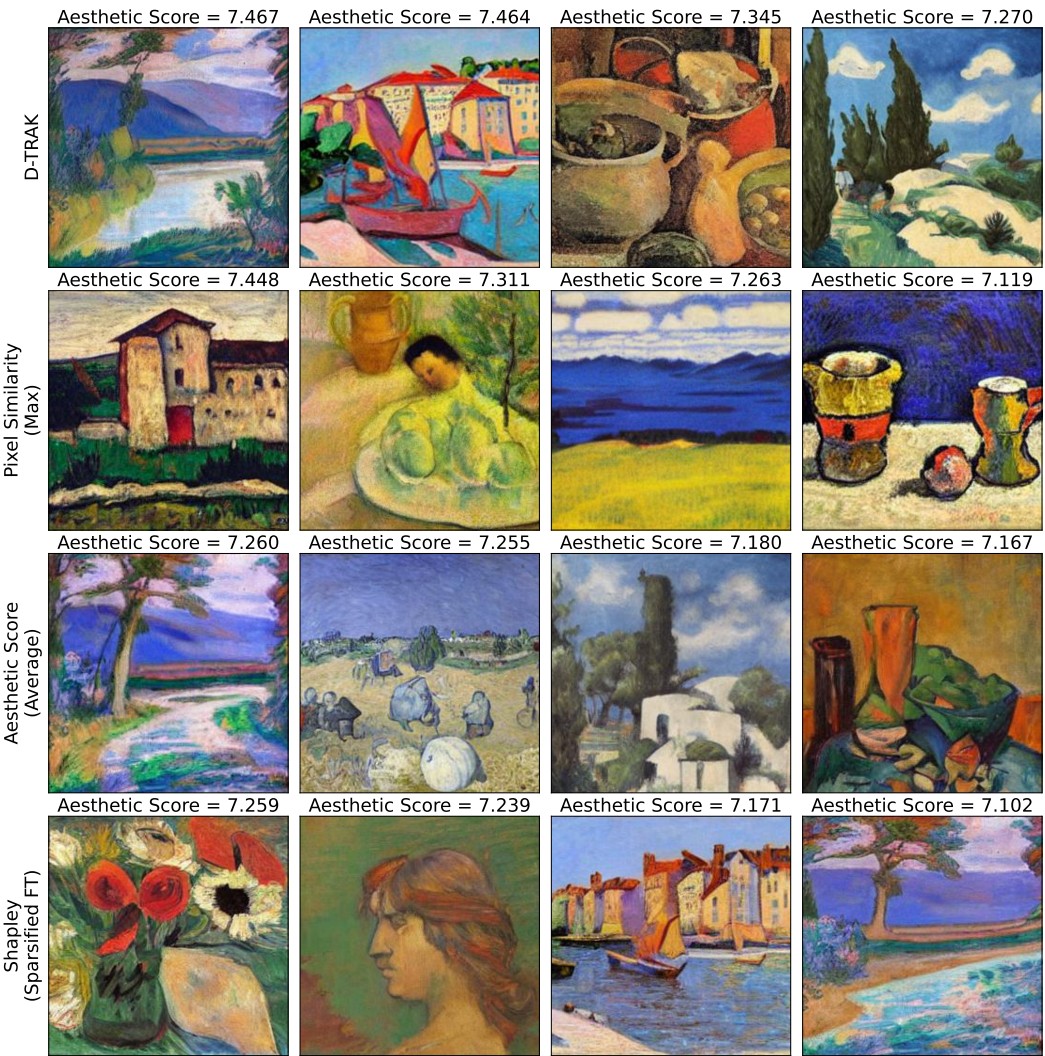

Figure 13: Two generated images above and two generated images below the 90th percentile of aesthetic scores, for Stable Diffusion models LoRA-finetuned without the top 40% most important artists based on D-TRAK, pixel similarity (max), training image aesthetic score (average), and sparsified-FT Shapley.

### F.4 ALTERNATIVE UNLEARNING APPROACHES FOR REMOVAL

In this section, we present the performance of unlearning approaches: fine-tuning (FT), gradient ascent (GA) (Graves et al., 2021), and influence unlearning (IU) (Izzo et al., 2021). To evaluate how well these approaches approximate retraining, we randomly selected 100 subsets from the Shapley sampling distribution, and we computed the FID score between the samples generated by the model retrained on each subset vs. models undergone by the unlearning methods. The resulting average FID scores are 25.5, 185.9, and 101.3 for sparsified fine-tuning (FT), sparsified GA, and sparsified IU, respectively (Table 12). In contrast, the average FID scores without sparsification are 43.1, 145.2, and 67.7 for fine-tuning (FT), GA, and IU, respectively (Table 12). These results indicate that while both GA and IU can effectively unlearn the target set, they significantly degrade overall image quality, Figure 14 and Figure 15. In contrast, sparsified FT demonstrates superior performance, producing better image quality than both GA and IU.

Table 12: FID scores between generated samples from different unlearning approximations and retrained models. Means and 95% confidence intervals across 100 random sampled subsets from Shapley distribution are reported.

| Unlearning Method | FID Score |
|---|---|
| *With Sparsification* | |
| Fine-Tuning (FT) | $25.5 \pm 10.5$ |
| Gradient Ascent (GA) | $185.9 \pm 45.7$ |
| Influence Unlearning (IU) | $101.3 \pm 32.1$ |
| *Without Sparsification* | |
| Fine-Tuning (FT) | $43.1 \pm 10.9$ |
| Gradient Ascent (GA) | $145.2 \pm 15.4$ |
| Influence Unlearning (IU) | $67.7 \pm 15.9$ |

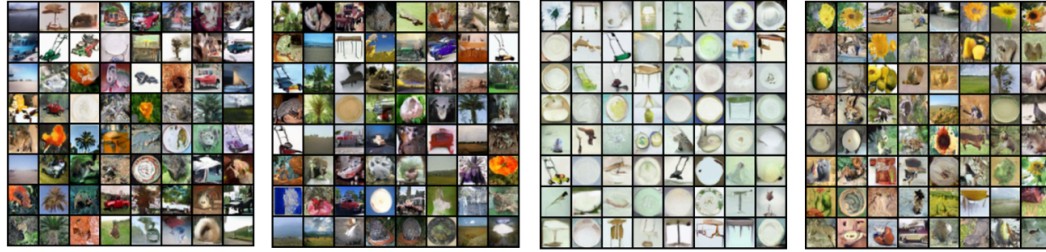

Figure 14: Sample images generated using various unlearning approaches on CIFAR-20 with identical noise inputs: retraining, sparsified fine-tuning (FT), sparsified gradient ascent (GA), and sparsified influence unlearning (IU) (from left to right).

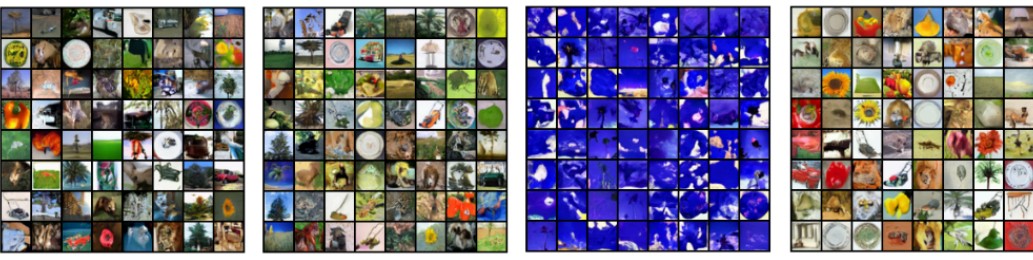

Figure 15: Sample images generated using various unlearning approaches on CIFAR-20 with identical noise inputs: retraining, fine-tuning (FT), gradient ascent (GA), and influence unlearning (IU) (from left to right).

### F.5 ANALYSIS OF TOP CONTRIBUTORS

Here, we present an additional analysis of the top contributors identified by sparsified FT. For CIFAR-20, we compute the entropy of the training images for each class based on the softmax outputs of InceptionNet. In the case of CelebA-HQ, the top contributors are predominantly drawn from non-majority clusters.

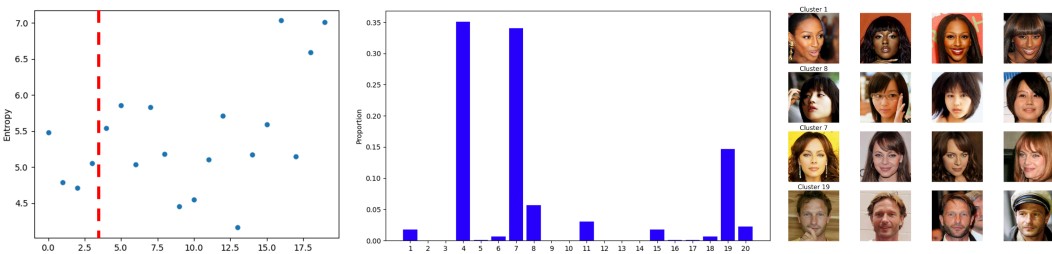

Figure 16: Entropy of each class in CIFAR-20 (left), redline indicates top 20% groups; Distribution of the demographic clusters in the CelebA-HQ dataset (middle). Top 4 celebrity identities by sparsified-FT Shapley (right).

## F.6 DATA QUALITY DISTRIBUTION

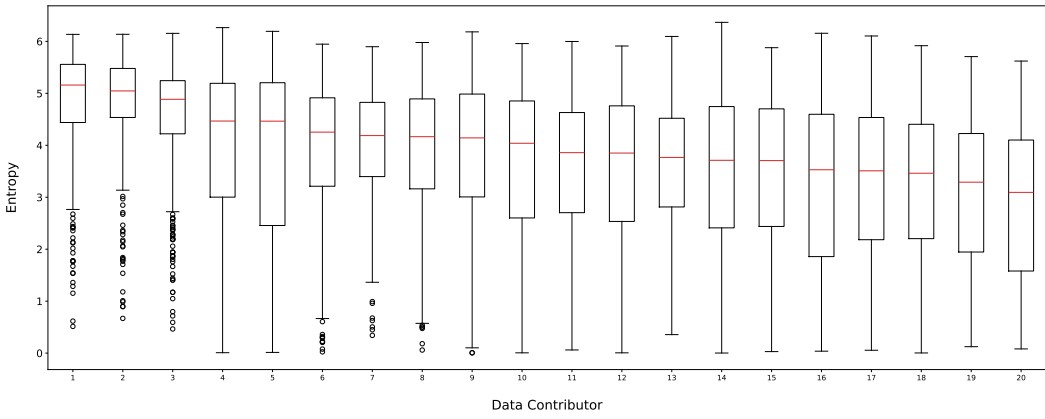

Figure 17: Entropy distributions across data contributors in CIFAR-20.

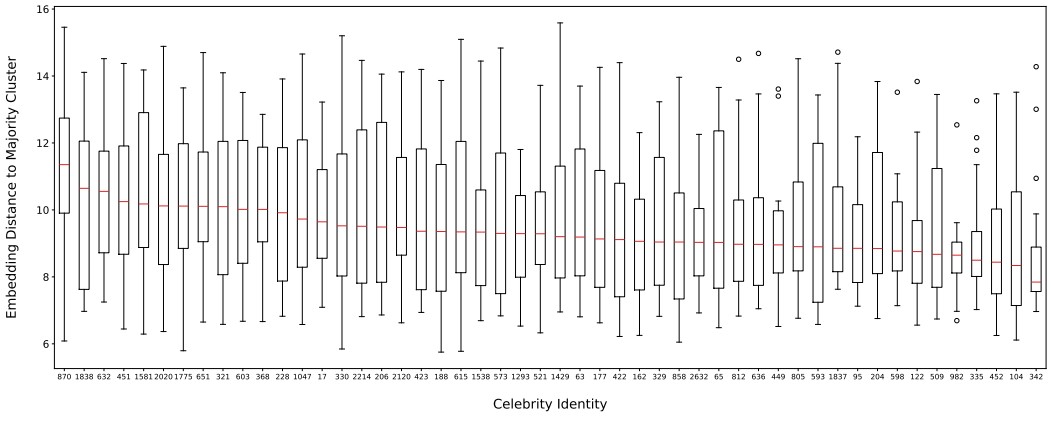

Figure 18: Distributions of embedding distance to the average of majority clusters across celebrity identities in CelebA-HQ.

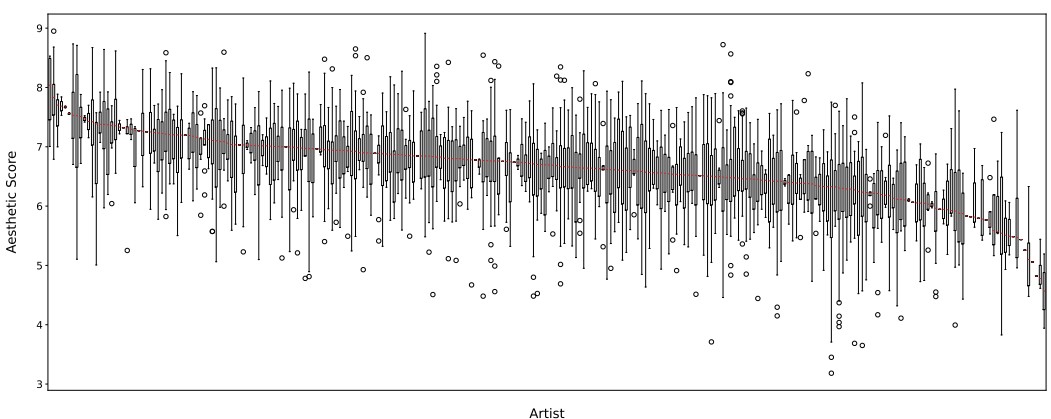

Figure 19: Distributions of aesthetic score across artists in ArtBench (Post-Impressionism).

