# OpenReview forum: "An Efficient Framework for Crediting Data Contributors of Diffusion Models"
_ICLR.cc/2025/Conference — ICLR 2025 Poster_

### Official Review · Reviewer_2xYg · 2024-11-02

**Soundness:** 2
**Presentation:** 2
**Contribution:** 2
**Rating:** 5
**Confidence:** 3

**Summary:**

This paper proposes to use a combination of kernel-based Shapley value and sparse fine-tuning as a new method to credit the data contributor in diffusion models. The authors evaluated their approach on CIFAR-20 CelebA-HQ and Art

**Strengths:**

If the comparison and contribution calculation methods are indeed correct and reasonable, the numerical results look good

**Weaknesses:**

I'm not very convinced about the overall proposed method for the following reasons.

For the sparsified FT part. some design choice can be elaborated and some comparisons with alternative methods can make the method more convincing.

1. Why is training on the full data and then finetuning on the subset comparable with training with subset from scratch?
2. Why pruning? Why not suing alternative solutions like full model with LoRA instead?
3. When applying sparsified FT, what are the contribution score formulas?

For the Shapley value part.

4. It will be better to elaborate on the similarity and differences between the conventional prediction tasks and the generation tasks, why the kernel based Shapley value can still work well in the current situation needs more explanation as well

For the numerical results, the ablation study is needed.

5. In table 1, the baselines used models trained with subset from scratch but the proposed method used sparsified FT. The results will be more straight forward when using retraining from scratch for all these scoring method or use sparsified FT for all these methods, that will show the ablation for each component in the new method

Overall speaking, the novelty seems fair.

6. The proposed method is a combination of Shapley value and sparse FT, and I think the reasoning for using this method will be much stronger if the authors can provide evidence showing that each component is better (either efficiency or effectiveness) than their alternatives in this task, like for contribution measurements, Shapley values vs LIME, PFI, etc., and for speeding up model retraining sparse FT vs unlearning, etc.

**Questions:**

Listed in the weaknesses

**Details Of Ethics Concerns:**

No concerns on this

---

> ### Author Response · Authors · 2024-11-23
>
> **Here we address the weaknesses raised point by point.**
>
> **1.** We thank the reviewer for raising this point to improve our paper. We have updated our manuscript to address this point with both theoretical and empirical results.
>
> In Proposition 1 of the revised manuscript, we show that the approximation error for Equation (6) is bounded when the number of sparsified fine-tuning steps increases asymptotically.  To summarize more formally, Proposition 1 shows that $E[|F(\tilde{\theta}^{\text{ft}}_{S, k}) - F(\theta^*_S)|] \le B$ for some constant $B > 0$ when the number of fine-tuning steps $k \to \infty$.
>
> In Proposition 2 of the revised manuscript, we show that the approximation error for Shapley values is also bounded when we increase the number of sparsified fine-tuning steps asymptotically. To summarize more formally, Proposition 2 shows that $E[\lVert\tilde{\beta}^{\text{ft}}_k - \beta^*\rVert_2] \le 2\sqrt{n} C$ for some constant $C > 0$ when $k \to \infty$. Here, $\tilde{\beta}^{\text{ft}}_k$ and $\beta^*$ denote Shapley values evaluated with global properties from sparsfied fine-tuned models and full-parameter models retrained from scratch, respectively.
>
> We also provide empirical results to assess the insights gained from our theoretical results (Propositions 1 and 2). As shown in Figure 5 in Appendix D of the revised manuscript, the approximation in Equation (6) empirically improves with more sparsified fine-tuning steps. As shown in Figure 6 in Appendix D of the revised manuscript, the similarity between the Shapley values estimated with retraining vs. sparsified fine-tuning indeed improves with more sparsified fine-tuning steps.
>
> **2.** We thank the reviewer for this comment. Since computing global model behaviors can require generating a large batch of samples (e.g., 10,240 generated images are required for the Inception Score in CIFAR-20), sparsification offers speed-ups for both unlearning and inference, making it a more suitable choice for our framework.
> While LoRA fine-tuning from a full model is a viable alternative, our results (Table 2 of Appendix F in the revised manuscript) show that although LoRA achieves a training speed-up comparable to sFT, its overall computational time remains longer than sFT due to the lack of inference speed-up. This highlights the advantage of sparsified fine-tuning.
> Additionally, we have experimented with using LoRA to calculate Shapley values on CIFAR-20 (Figure 8 of Appendix F in the revised manuscript) and CelebA-HQ (Figure 10 of Appendix F in the revised manuscript). The results demonstrate that sFT consistently outperforms LoRA under the same computational budget.
> Table 2 Average runtime for data subset in minutes (training + inference) for Shapley values estimated with retraining, fine-tuning(FT), sparsified FT, and LoRA fine-tuning
> | **Method** | **CIFAR-20**       | **CelebA-HQ**     | **Artbench (Post-Impressionism)** |
> |------------|--------------------|-------------------|-----------------------------------|
> | Retrain    | 77.4 ± 20.8        | 213.4 ± 26.5      | 190.6 ± 6.4                      |
> | FT         | 6.06 ± 18.9        | 17.5 ± 27.0       | 4.4 ± 6.4                        |
> | sFT        | 4.37 ± 14.0        | 11.0 ± 11.9       | 4.5 ± 6.1                        |
> | LoRA       | 4.21 ± 19.0        | 5.5 ± 26.7        | —                                |
>
> **3.** To calculate sparsified-FT Shapley values, $F(\theta^*_{S_j})$ (which denotes the global property evaluated with a full-parameter model retrained on the subset $S_j$) in Equation (5) is replaced with $F(\tilde{\theta}^{\text{ft}}_{S_j, k})$ (which denotes the global property evaluated with a pruned model fine-tuned on the same subset with $k$ steps).
> Concretely, if we introduce the shorthand notations:
>
> $A = \frac{1}{M} \sum_{j=1}^M 1_{S_j} 1_{S_j}^T$
>
> and
>
> $b = \frac{1}{M} \sum_{j=1}^M 1_{S_j} (F(\tilde{\theta}^{\text{ft}}_{S_j, k}))$
>
> $c = \frac{1}{M} \sum_{j=1}^M  1_{S_j}( F(\theta_{\emptyset}))$
>
> Then the Shapley values have the closed-form expression:
>
> $\beta = A^{-1} \left( (b -c) - 1 \frac{1^T A^{-1} (b -c) - F(\theta^*) + F(\theta_{\emptyset})}{1^T A^{-1} 1} \right)$.

---

> ### Author Response · Authors · 2024-11-23
>
> **4.** We thank the reviewer for raising this question. The Shapley value is a general game-theoretic framework that works across various scenarios, as it only requires sampling subsets of "players" (data contributors in this context) and evaluating their impact on a given model behavior. This flexibility allows the Shapley value to be adapted for different task settings, including both conventional prediction tasks and generative tasks.
> The key difference between these tasks lies in how model behavior is defined. In prediction tasks, model behavior is typically captured by each dimension of model output, such as class probabilities, or by using metrics like classification accuracy, precision, or recall. In contrast, generative tasks present unique challenges: the generation process is inherently stochastic (as models produce different outputs based on varying initial Gaussian noise inputs) and the outputs are high-dimensional (e.g., 224 × 224 = 50,176 pixels), with individual dimensions lacking direct semantic meaning.
> To address these challenges, we propose defining model behavior in terms of global model properties. Specifically, we evaluate a batch of generated data using application-relevant metrics that capture key aspects of generated outputs, such as image quality and diversity. This enables us to align the Shapley value framework with generative tasks, and we show it remains effective via quantitative and qualitative evaluations.
>
> **5.** We thank the reviewer for suggesting this ablation study. We have conducted an analysis summarizing the results of retraining-based attribution methods from existing works, including the Shapley value, the Banzhaf value, and Leave-One-Out (LOO), evaluated under three scenarios: (1) retraining from scratch, (2) fine-tuning (FT), and (3) sparsified fine-tuning (sFT). The following table presents the results for CIFAR-20.
>
> Table 5 of Appendix F.1: LDS (%) results for retraining-based attribution across Shapley, Leave-One-Out (LOO), and Banzhaf distributions with α = 0.25, 0.5, 0.75 on CIFAR-20. The global model behavior is evaluated using the Inception Score of 10,240 generated images. Means and 95% confidence intervals across three random initializations are reported. For sparsified fine-tuning (sFT) and fine-tuning (FT), the number of fine-tuning steps is set to 1000.
>
> | **Method**          | **α = 0.25**         | **α = 0.5**          | **α = 0.75**         |
> |---------------------|----------------------|-----------------------|----------------------|
> | LOO (retraining)    | 17.01 ± 5.29         | 30.66 ± 6.11          | 13.64 ± 4.99         |
> | Banzhaf (retraining)| 10.59 ± 8.64         | 37.11 ± 2.33          | 42.78 ± 1.85         |
> | Shapley (retraining)| **65.90 ± 1.52**     | **70.58 ± 2.05**      | **72.07 ± 4.83**     |
> |---------------------|----------------------|-----------------------|----------------------|
> | LOO (FT)            | -55.00 ± 11.76      | -66.06 ± 2.28         | -54.58 ± 7.02        |
> | Banzhaf (FT)        | -11.20 ± 6.42       | 9.09 ± 2.67           | 16.79 ± 1.12         |
> | Shapley (FT)        | **20.57 ± 3.02**    | **39.60 ± 3.03**      | **38.51 ± 4.62**     |
> |---------------------|----------------------|-----------------------|----------------------|
> | LOO (sFT)           | 29.45 ± 5.96        | 27.43 ± 4.20          | 19.58 ± 0.35         |
> | Banzhaf (sFT)       | 5.44 ± 8.59         | 22.55 ± 5.07          | 31.44 ± 0.27         |
> | Shapley (sFT)       | **51.24 ± 3.39**    | **61.48 ± 2.27**      | **59.15 ± 4.24**     |
>
> Our findings across datasets demonstrate that Shapley value consistently outperforms both LOO and Banzhaf value across all training scenarios. Among the training procedures, retraining (exact unlearning) from scratch expectedly delivers the best performance across all three attribution methods. For unlearning approximation, sparsified fine-tuning (sFT) achieves superior performance compared to fine-tuning (FT), highlighting its effectiveness within our framework. Please refer to the updated manuscript for the results of CelebA-HQ (Table 8 of Appendix F.2) and ArtBench (Table 11 of Appendix F.3).

---

> ### Author Response · Authors · 2024-11-23
>
> **6.** We thank the reviewer for this comment. As suggested, we have also compared different retraining approximations, including fine-tuning (FT), gradient ascent (GA), influence unlearning (IU), and their sparsified version, and show that sFT provides the best performance. For GA and IU, we found that their retraining approximations, both with and without sparsification, were ineffective (refer to Table 12 in Appendix F.4).
>
> Building on this, we conducted an ablation study across different attribution kernels—Shapley, leave-one-out (LOO), and Banzhaf—evaluated using FT, sFT, and retraining (Tables 5, 8, and 11 in Appendix). Among these kernels, the Shapley value consistently demonstrated superior performance. Furthermore, within the Shapley framework, sFT outperformed other retraining approximations when operating under the same computational budget.

---

> > ### Author Response · Authors · 2024-11-30
> >
> > We appreciate the thoughtful comments provided by the reviewer. We hope our responses have adequately addressed your concerns. Should you have any further questions or need additional clarification, please do not hesitate to let us know, and we will address them promptly.

---

### Official Review · Reviewer_f56T · 2024-11-03

**Soundness:** 3
**Presentation:** 3
**Contribution:** 3
**Rating:** 8
**Confidence:** 4

**Summary:**

The paper entitled "An Efficient Framework for Crediting Data Contributors of Diffusion Models" has focused on diffusion models and presented a method to fairly attribute data contributions using Shapley values. To address computational inefficiencies, the authors employ model pruning and fine-tuning, enabling practical Shapley value estimation. Their method is validated across multiple datasets, demonstrating improved attribution accuracy and efficiency over existing techniques.

**Strengths:**

1- The Shapley theorem has been proven to be an effective solution for calculating the contribution and is widely used in data valuation. The author has properly utilised this theorem as part of the methodology.
2- Regarding quality, the methodology is well-executed, with rigorous evaluations across multiple datasets that demonstrate the proposed framework’s superior performance.
3- About the clarity, the paper is well-written, with structured explanations and nice visualization,

**Weaknesses:**

1- It is not a weakness, but the author could also provide an evaluation against data poisoning, which could make it even stronger.
2- The provided code is well-structured but it could be improved by providing further demo Jupyter notebooks that make it easier for others to test and run the model.

**Questions:**

How does the framework handle cases where data contributors have varying data quality, styles, etc. and could this affect the accuracy of attribution?

---

> ### Author Response · Authors · 2024-11-23
>
> **Here we address the weaknesses raised point by point.**
>
> **1.** We thank the reviewer for this suggestion while recognizing that this is not a weakness of our paper. Indeed, data attribution methods for supervised models have been shown capable of detecting data poisoning, while data attribution methods for diffusion models have not focused on this capability [R1-R2]. We believe a comprehensive evaluation against data poisoning across different data attribution methods for diffusion models would be an impactful work and leave that for future research.
>
> [R1] Zheng et al. - Intriguing Properties of Data Attribution on Diffusion Models
>
> [R2] Georgiev et al. - The Journey, Not the Destination: How Data Guides Diffusion Models
>
> **2.** We thank the reviewer for this valuable suggestion! In response, we have included a tutorial.md file to provide more detailed instructions for replicating our experiments, including steps for retraining, various unlearning methods, and LDS evaluation. We hope this will make it easier for others to test and utilize our method.
>
> **Here we address the questions raised one by one.**
>
> **1.** We thank the reviewer for this insightful question. Varying data quality can be an important factor to consider when attributing credit. Our experiments take this into account because there is inherent variability in the datasets. In the updated manuscript, we show variations in image quality within and across data contributors. In Figure 17 of Appendix F.6, the entropy distribution for each data contributor’s images is shown. For example, while the first data contributor shows a high median entropy, there are individual images with low entropy, indicating varying data quality within the contributor's provided data. Similarly, in CelebA-HQ (Figure 18 of Appendix F.6), when measuring the embedding distance to the majority cluster for images of each celebrity, we observe significant variability in data quality both within a single celebrity’s images and across different celebrities. In Figure 19 of Appendix F.6, inter- and intra-artist variations in image aesthetic scores are also observed for ArtBench (Post-Impression). Given these variations, our experiment results show that our framework can still perform well.

---

> > ### Author Response · Authors · 2024-11-30
> >
> > We appreciate the thoughtful comments provided by the reviewer. We hope our responses have adequately addressed your concerns. Should you have any further questions or need additional clarification, please do not hesitate to let us know, and we will address them promptly.

---

### Official Review · Reviewer_KRfe · 2024-11-06

**Soundness:** 3
**Presentation:** 3
**Contribution:** 2
**Rating:** 5
**Confidence:** 3

**Summary:**

This paper presents a framework for attributing the contributions of data providers in diffusion models. The authors propose a framework that efficiently approximates retraining and rerunning inference for diffusion models, thus enabling the estimation of Shapley values for data contributors. This is achieved by investigating how global properties of diffusion models are influenced by data contributors. Empirically, it is demonstrated that the proposed framework outperforms existing data attribution
methods across three datasets, model architectures, and global properties.

**Strengths:**

Overall, I think this paper is organized and written clearly and I enjoyed reading most of the parts. In particular, the conceptual strengths of this paper include:
1. The proposed framework utilizes Shapley values, a game-theoretic approach, to assign fair credit to data contributors based on their influence on model performance. This methodology uniquely meets the fairness principles in valuation.
2. To address the high computational cost of Shapley value calculations, the paper introduces a model-pruning and fine-tuning method, which significantly accelerates retraining and inference processes.
3. The approach has the potential to be applicable in various scenarios, such as incentivizing quality data sharing, creating compensation policies, and improving model diversity and fairness, making it a nice tool for real-world diffusion model deployments​.

**Weaknesses:**

1. despite the computational efficiency of the proposed speed up method with sparsified fine-tuning (section 3.2), it seems to me that there is no in-depth discussion about its accuracy. The core idea of the approximation is Eq. (6), but there is no discussion or empirical evidence to support how well the approximation (6) is. Given that the idea is straightforward, it would be much more convincing if the author can provide additional justifications for Eq. (6), besides its superior empirical performance compared to baseline methods. Otherwise, it is difficult to digest why the proposed approach outperforms other methods with such a dominant advantage (Table 1).
2. I do not see any particular reason the approximation for computing Shapley value has to be restricted in the diffusion model applications. Whether this is true or not, it would be better to include additional discussions in this regard.
3. I feel the contributions summarized at the end of the introduction is a bit over-claimed. For example, the first claimed contribution is not surprising from my perspective, as it is well-acknowledged that the performance of any ML model relies heavily on the sources of its training data set. For the second claim, I'm not so sure what does "efficiently approximate" mean. From the paper I get that the proposed approximation framework indeed reduces the computational efficiency of solving the least square problem (5), however, there is no evidence of how well the approximation is. In my opinion, an "efficient" approximation should somehow provide a trade-off between computational cost and approximation accuracy. That said, an approximation method with only a computational cost guarantee makes it less convincing and lacks insight. I think this paper can benefit more from an in-depth discussion of the proposed approximation approach.
4. the focus of technical writing is not well-balanced. In my opinion, the entire section 2 and section 3.1 are known results (which do not contribute to the novelty of this work) and should be shortened significantly. However, unfortunately, the core novel part of the proposed method (section 3.2) is not discussed in depth.

**Questions:**

1. in definition 1, you said the function $\tau(\mathcal{F}, \\{C_i\\}_{i=1}^n)$ is supposed to assign a score to each contributor $i$. I'm wondering how this notation reflects this idea. Maybe it's a typo and it should be $\tau(\mathcal{F}, C_i)$?
2. if I understand it correctly, in the experiment results (section 4.5), the LDS is computed with different baseline methods for computing $\tau$. Then how are $\{\mathcal{F}(\theta^*_{S_b})\}_{b=1}^B$ computed? Are they computed by Sparsified-FT Shapley? If this is the case, why it is a fair comparison if the proposed approach is served as the benchmark in the evaluation metric?
3. is the proposed approach applicable to any data-driven machine learning model? I don't see any reason why it must be restricted to the diffusion model. If this is the case, why this paper focus on the application of diffusion models?

---

> ### Author Response · Authors · 2024-11-23
>
> **Here we address the weaknesses raised point by point.**
>
> **1.** We thank the reviewer for raising this point to improve our paper. We have updated our manuscript to address this point with both theoretical and empirical results.
>
> In Proposition 1 of the revised manuscript, we show that the approximation error for Equation (6) is bounded when the number of sparsified fine-tuning steps increases asymptotically.  To summarize more formally, Proposition 1 shows that $E[|F(\tilde{\theta}^{\text{ft}}_{S, k}) - F(\theta^*_S)|] \le B$ for some constant $B > 0$ when the number of fine-tuning steps $k \to \infty$. In Proposition 2 of the revised manuscript, we show that the approximation error for Shapley values is also bounded when we increase the number of sparsified fine-tuning steps asymptotically. To summarize more formally, Proposition 2 shows that $E[\lVert\tilde{\beta}^{\text{ft}}_k - \beta^*\rVert_2] \le 2\sqrt{n} C$ for some constant $C > 0$ when $k \to \infty$. Here, $\tilde{\beta}^{\text{ft}}_k$ and $\beta^*$ denote Shapley values evaluated with model properties from sparsfied fine-tuned models and fully retrained models, respectively.
>
> We also provide empirical results to assess the insights gained from our theoretical results (Propositions 1 and 2). As shown in Figure 5 in Appendix D of the revised manuscript, the approximation in Equation (6) empirically improves with more sparsified fine-tuning steps. As shown in Figure 6 in Appendix D of the revised manuscript, the similarity between the Shapley values estimated with retraining vs. sparsified fine-tuning indeed improves with more sparsified fine-tuning steps.
>
> **2.** We appreciate the reviewer for giving us the opportunity to elaborate on this point, as well as on the related Question 3.  You are correct that the proposed approach for accelerating the computation of Shapley values for data contributors is not inherently restricted to diffusion models and could be broadly applied to other large-scale deep learning models. We chose to focus on diffusion models for this paper as they are state-of-the-art models for generating realistic images and they have been widely used for real-world applications, including artwork generation. This has sparked widespread discussions around issues such as data attribution and royalties, making diffusion models an ideal testbed for our framework. This targeted focus allowed us to conduct rigorous experiments across diverse datasets, demonstrating the utility and efficiency of our approach.
> That said, we agree that the methodology could be extended to other models, such as large language models, where similar computational challenges arise. In such cases, appropriate global model properties would need to be defined, and parameters for sparsified finetuning would require tuning to optimize the retraining and inference process. We believe extending our method to other model types is indeed an exciting avenue for future research, and we have noted this in the revised manuscript as a potential avenue for further exploration.
>
> **3.** We thank the reviewer for the feedback. To clarify our contribution in the updated manuscript, we rephrase the first claim to: "we are the first to investigate how to attribute global properties of diffusion models to data contributors.” This rephrasing aims to clarify that our contribution focuses on crediting data contributors instead of assessing how data affect model performance.
> In the updated manuscript, we include both theoretical and empirical results to demonstrate that our method with sparsified fine-tuning can approximate Shapley values estimated via retraining. In Proposition 2 of the updated manuscript, we show that the approximation error for Shapley values is bounded when the number of sparsified fine-tuning steps increases asymptotically. In Figure 6 in Appendix D of the updated manuscript, it is empirically shown that increasing the number of sparsified fine-tuning steps indeed corresponds to improved similarity between the Shapley values estimated via retraining vs. sparsified fine-tuning. In Figure 7 in Appendix D of the updated manuscript, it is shown that the average runtime increases linearly with the number of sparsified fine-tuning steps. With Figures 6 and 7, the empirical trade-off between approximation accuracy and computational cost are shown. With these theoretical and empirical results, the second contribution claimed in the introduction is now more substantiated.
>
> **4.** We thank the reviewer for this suggestion to improve our paper. In the updated manuscript, Section 2 is shortened to remove non-essential details. More importantly, Section 3.2 is now updated to include an in-depth discussion with theoretical justifications (Propositions 1 and 2) for sparsified fine-tuning.

---

> > ### Author Response · Authors · 2024-11-23
> >
> > **Here we address the questions raised one by one.**
> >
> > **1.** We thank the reviewer for pointing out the potential confusion in our writing. To clarify, the output of $\tau$ is $n$-dimensional, with the $i$th element corresponding to the attribution score for the $i$th data contributor. In the updated manuscript, we rephrase the last sentence in Definition 1 to: “A contributor attribution method is a function $\tau(F, \\{C_i\\}_{i=1}^n)$ that assigns scores to all contributors to indicate each contributor’s importance to the global model property $F$.”
> >
> > **2.** We thank the reviewer for raising this question. To clarify, the LDS evaluation uses $\\{F(θ_{S_b}^∗)\\}_{b=1}^B$ as oracle ground truths, which are computed through **exact retraining**, rather than relying on any unlearning approximation. The subsets $S_b$ are sampled from the **datamodel distribution** (random subsets with $\alpha \cdot n$ data contributors), which is different from the Shapley kernel distribution. Hence, sparsified-FT Shapley does not have an unfair advantage with respect to the LDS evaluation.

---

> > > ### Author Response · Authors · 2024-11-30
> > >
> > > We appreciate the thoughtful comments provided by the reviewer. We hope our responses have adequately addressed your concerns. Should you have any further questions or need additional clarification, please do not hesitate to let us know, and we will address them promptly.

---

> > ### Comment · Reviewer_KRfe · 2024-12-03
> >
> > I appreciate the authors' detailed response. However, I believe the new results do not fully address my main concern. The current error bounds lack sufficient insight: our understanding of the constants $B$ and $C$ in the bounds is limited, which prevents us from drawing meaningful conclusions about the accuracy of the approximation based on these bounds.
> >
> > I am not insisting that the paper must include concrete theoretical results. My main concern is that the work currently lacks strength in both theoretical and empirical contributions. If the authors intend to present this as a theory paper, it would be acceptable to focus on a particular or even simple subset of models. However, it is crucial to derive theoretical results that offer new insights. In its current form, the theoretical results, particularly Propositions 1 and 2, are not compelling enough. I would encourage the authors to provide further explanation or context to help readers better appreciate these results.
> >
> > If the authors aim to position this as an empirical work, that approach is also perfectly valid. However, I would then expect the solution design to be more specifically tailored to the unique features of diffusion models. As it stands, the work appears to incrementally apply the idea of Shapley values to the training of diffusion models. Moreover, the authors acknowledge that their solution is not limited to diffusion models, which raises another concern: if the solution is not tailored to diffusion models, the emphasis should instead be on the generalizability of the approach, preferably demonstrated across a broader range of models. Unfortunately, this aspect is also underexplored in the current submission.
> >
> > Therefore, I suggest three potential ways to improve the paper:
> >
> > - Focus on the theoretical aspect: Derive why the computation of Shapley values can be efficiently and approximately computed for a specific class of models, thus providing theoretical insights that can potentially impact both algorithmic game theory and applied ML communities.
> >
> > - Focus on the empirical side: Propose a solution that leverages the unique features or structures of diffusion models and support it with experiments that demonstrate its effectiveness.
> >
> > - Still focus on the empirical side: Test the currently proposed solution on a wide range of popular models to showcase its generalizability and applicability across different settings.
> >
> > I would be happy to champion the paper if it is organized around any of these narratives. However, in its current form, I will maintain my borderline rating.

---

> > > ### Author Response · Authors · 2024-12-04
> > >
> > > While we appreciate the reviewer’s response to our rebuttal, the response presents a narrow view on what constitutes meaningful contributions and undervalues the main aspects of our paper’s contribution.
> > >
> > > Our theoretical results provide insights into the role of the number of fine-tuning steps $k$, suggesting that $k$ should be as large as possible within a computational budget. This insight is further substantiated by empirical results in Figures 5 and 6 in Appendix D of the revised manuscript. Regarding our empirical contribution, we focus on the timely and pressing problem of attributing data contributors for diffusion models, which “has the potential to be applicable in various scenarios, such as incentivizing quality data sharing, creating compensation policies, and improving model diversity and fairness, making it a nice tool for real-world diffusion model deployments​ [Strength 3 mentioned by the same reviewer].” With respect to the problem of attributing data contributors for diffusion models, our approach empirically performs the best by large margins (e.g., ~30.8%, ~4.6%, ~36.6% LDS for CIFAR-20, CelebA-HQ, and ArtBench Post-Impressionism, respectively as shown in Table 1). It is surprising that such performance improvements for an important problem are not considered a strong empirical contribution.
> > >
> > > Also, we respectfully disagree with the notion that focusing on a specific, important model type is a weakness of our paper. The primary goal of our work is to address the pressing challenge of efficient data attribution for diffusion models, rather than creating a universally applicable method—though our framework may have broader applicability. Previous studies on data valuation have similarly focused on specific model types (i.e., supervised models), despite their theoretical applicability to other settings, such as unsupervised models like VAEs [R3-R5]. We focused on diffusion models due to pressing needs, and this presented unique computational challenges. Unlike supervised models, where retraining for data valuation is computationally feasible [R6], retraining diffusion models for data valuation was previously impractical. The reviewer’s view that applying the idea of Shapley values to diffusion models is incremental overlooks the significant computational barriers addressed in our work. To the best of our knowledge, our work is the first to make Shapley value estimation computationally practical (i.e., completed under one day with 8 RTX-6000 GPUs) for diffusion models.
> > >
> > > References
> > >
> > > [R3] Ghorbani et al. - What is your data worth? Equitable Valuation of Data
> > >
> > > [R4] Kwon et al. - Beta Shapley: a Unified and Noise-reduced Data Valuation Framework for Machine Learning
> > >
> > > [R5] Wang et al. - Data Banzhaf: A Robust Data Valuation Framework for Machine Learning
> > >
> > > [R6] Ilyas et al. - Datamodels: Predicting Predictions from Training Data

---

### Author Response · Authors · 2024-11-28
**Summary of revision**

We thank the reviewers for reviewing our paper and for providing thoughtful and constructive feedback. We are pleased to see that the reviewers recognize the importance of the problem of crediting data contributors and acknowledge our proposed approach using the Shapley value and its adaptation to real-world scenarios [KRfe, f56T].

In response to the reviewers' questions and concerns, we have provided clarifications, introduced theoretical results, and performed additional experiments, all of which are detailed in the individual responses. To summarize, we conducted theoretical analysis on the approximation error of sparsified fine-tuning and validated the insights through empirical experiments [KRfe, 2xYg]. These results have been incorporated into the exposition of our method (Section 3.2) in the revised manuscript, providing a more comprehensive explanation of our approach [KRfe]. To demonstrate the effectiveness of the Shapley kernel distribution, we conducted additional ablation studies comparing against other distributions, such as leave-one-out (LOO) and Banzhaf (Tables 5, 8, and 11 in the revised manuscript) [2xYg]. We also performed an analysis of the data distribution to highlight variations in image quality both within and across different data contributors, further showcasing the applicability of our approach [f56T]. Furthermore, to demonstrate the efficiency of sparsified fine-tuning, we included another unlearning approximation--fine-tuning with LoRA--and evaluated LDS under the same computational budgets (Figures 8 and 10 in the revised manuscript) [2xYg]. Finally, we clarified our motivation for focusing on diffusion models in this paper, due to their widespread use in real-world applications, and the pressing need to address issues such as royalty and credit attribution for data contributors [KRfe]. We believe that our proposed approach has broader applicability and could have significant impact on other data-driven models and scenarios where accurate data attribution is needed, leaving that for future work.

For more details, please refer to our responses to individual reviewers and the revised manuscript. We believe our responses comprehensively address the reviewers’ concerns and ensure these clarifications and additional results are included in our revised manuscript. We look forward to your response and are happy to address any further questions.

---

### Meta-Review · Area_Chair_Dejg · 2024-12-23

**Metareview:**

This paper looks at data attribution in trained diffusion model through the lens of Shapley values, a common and accepted method originally developed in the economics literature to credit agents’ varying contributions in a cooperative game and has more recently been applied in feature-based attribution methods such as SHAP in XAI, and in data value assignment in settings such as the present paper.  The proposed approach, roughly stated, takes a trained diffusion model, distills/prunes it in some way, and then fine-tunes a set of these small models specifically on subsets of data belonging to an individual contribution.  At inference time, it’s able to scalably estimate Shapley values by hitting subsystems built around these fine-tuned models.  Reviewers appreciated the focus of the paper and its motivation.

**Additional Comments On Reviewer Discussion:**

Some reviewers (e.g., KRfe) brought up concerns over the theoretical results’ useful, which this AC lightly shares; that said, to my knowledge this is the first even roughly scalable approach to Shapley-value-based attribution/value problems in diffusion models, which is important in its own right.  The other 5-scoring reviewer 2xYg did not participate in the rebuttal process; looking at their review and the extensive rebuttal, it’s this AC’s opinion that the new theoretical clarifications & experimental results address many of their concerns.

---

### Decision · Program_Chairs · 2025-01-22

Accept (Poster)